# Diffuse Everything: Multimodal Diffusion Models on Arbitrary State Spaces

Kevin Rojas [* 1 2]   Yuchen Zhu [* 1 2]   Sichen Zhu [2]   Felix X.-F. Ye [3]   Molei Tao [1 2]

## Abstract

Diffusion models have demonstrated remarkable performance in generating unimodal data across various tasks, including image, video, and text generation. On the contrary, the joint generation of multimodal data through diffusion models is still in the early stages of exploration. Existing approaches heavily rely on external preprocessing protocols, such as tokenizers and variational autoencoders, to harmonize varied data representations into a unified, unimodal format. This process heavily demands the high accuracy of encoders and decoders, which can be problematic for applications with limited data. To lift this restriction, we propose a novel framework for building multimodal diffusion models on arbitrary state spaces, enabling native generation of coupled data across different modalities. By introducing an innovative decoupled noise schedule for each modality, we enable both unconditional and modality-conditioned generation within a single model simultaneously. We empirically validate our approach for text-image generation and mixed-type tabular data synthesis, demonstrating that it achieves competitive performance. Code is available at Diffuse-Everything.

## 1. Introduction

Recent years have witnessed the tremendous success of diffusion generative models in various applications. The seminal works of continuous diffusion models on Euclidean spaces (Sohl-Dickstein et al., 2015; Song et al., 2020; Ho et al., 2020) have led to state-of-the-art methods for tasks such as image generation (Dhariwal & Nichol, 2021; Bao et al., 2023a; Karras et al., 2022; 2024b), video generation

(Ho et al., 2022; Jin et al., 2025), time series forecasting (Chen et al., 2024b; Rojas et al., 2025b) and in domains such as robotics (Chi et al., 2023) and genomics (Luo et al., 2024; Zhu et al., 2025a). Pioneering works have shown that diffusion models can also be extended to curved spaces (De Bortoli et al., 2022; Huang et al., 2022; Cheng et al., 2025; Chen & Lipman, 2024; Zhu et al., 2025b), enabling a high-fidelity generation of structured data on manifolds, such as material configurations (Sriram et al., 2024) and protein backbones (Watson et al., 2023; Yim et al., 2023). Recently, discrete diffusion models have emerged as the cornerstone for modeling categorical data with inherent discrete structures (Campbell et al., 2022; Lou et al., 2024; Campbell et al., 2024; Gat et al., 2024). Discrete diffusion models have imposed great impacts on protein sciences (Wang et al., 2024b), graph generation (Xu et al., 2024; Li et al., 2025), and text generation (Nie et al., 2025a;b). In general, diffusion models have shown top performance in most scenarios with unimodal data.

Generative models also demonstrated successes in multimodal data lately. For example, conditional diffusion models showed remarkable capabilities in tasks such as text-to-image generation by accurately synthesizing pictures following given instruction prompts (Ramesh et al., 2022; Chen et al., 2023; Esser et al., 2024). It's worth noting that such models still generate single-modality outputs (such as images). Therefore, to jointly generate multimodal data, leveraging only single-task-performing conditional models is extremely computationally inefficient, as it requires combining **multiple** independently trained models by sequentially applying them.

An alternative approach is to use a **single** multi-modal model that captures the joint distribution of multiple modalities. Such an approach often leads to strong performances as it allows information to mix across modalities (Li et al., 2024a; Meta, 2024). Existing approaches of this type are mainly based on autoregressive models (AR), such as Chameleon (Meta, 2024) and Unified-IO (Lu et al., 2024), where data of different modalities are represented uniformly as tokens and generated autoregressively from left to right. Apart from these, attempts have also been made to realize this idea using diffusion/flow-based methods, such as UniDiffuser (Bao et al., 2023b), MM-Diffusion (Ruan et al., 2023), AVDiT (Kim et al., 2024), UniDisc (Swerdlow et al., 2025),

[*]Equal contribution, random order by coin flip [1]School of Mathematics, Georgia Institute of Technology, Atlanta, GA [2]Machine Learning Center, Georgia Institute of Technology, Atlanta, GA [3]Department of Mathematics & Statistics, SUNY Albany, NY. Correspondence to: Molei Tao <mtao@gatech.edu>.

*Proceedings of the $42^{nd}$ International Conference on Machine Learning*, Vancouver, Canada. PMLR 267, 2025. Copyright 2025 by the author(s).

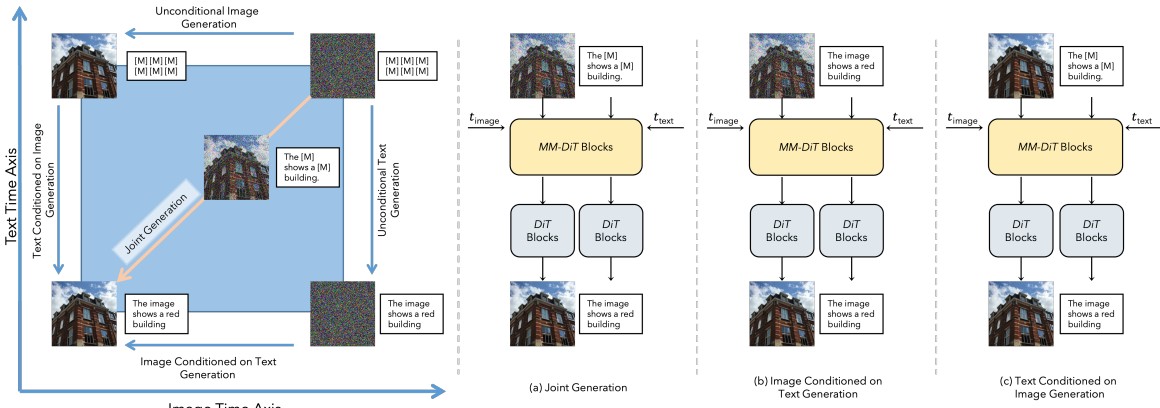

*Figure 1.* By injecting noise into different modalities in a decoupled fashion, we enable the unconditional and modality-conditioned generation in a single model. (a) Joint generation of image and text. (b) Image generation given text captions as conditions. (c) Text generation given images as conditions.

OmniFlow (Li et al., 2024a), etc. These methods generate multimodal data simultaneously through iterative denoising of a randomly sampled initial noise.

A commonality among the aforementioned methods designed for joint multimodal generation is that they typically rely heavily on preprocessing techniques to **harmonize the varied data representations into a unified format**, thereby creating a single modality. One approach (taken by Chameleon (Meta, 2024), UniDisc (Swerdlow et al., 2025), etc) is to cast multimodal inputs all into discrete tokens with modality-dependent tokenizers built with discrete or vector-quantized variational encoders (VQVAE) (Van Den Oord et al., 2017; Esser et al., 2021). An alternative route (considered by UniDiffuser (Bao et al., 2023b), OmniFlow (Li et al., 2024a), etc) is to preprocess the multimodal data by embedding them into continuous-valued latent vectors with encoders trained with variational encoders (VAE) (Kingma, 2013) or representation alignment (e.g., CLIP (Radford et al., 2021)).

For these approaches, regardless of whether discrete tokens or continuous latents are used, generation is performed in a unimodal space, and the original data modality must be recovered through decoding. Therefore, these pipelines may suffer from generation artifacts due to the limited accuracy of the decoders (Hoogeboom et al., 2024). Additionally, the requirement for high-performance encoder-decoder pairs can be problematic to satisfy for applications that lack abundant high-quality data (Zhang et al., 2023). Due to the requirement for task-specific algorithm designs, these methods also cannot be conveniently extended to generate data composed of arbitrary modalities. Therefore, a natural question to ask is the following:

*Can we design a principled framework to enable joint modeling of multi-modal data in their native spaces without a unified representation?*

Diffusion models serve as a powerful backbone for building such a framework. Theorists have shown that diffusion models can be extended to a more general idea called *denoising Markov models* (Benton et al., 2024; Ren et al., 2025b), providing a solid theoretical foundation for a **multimodal extension**. In addition to this, existing works such as MultiFlow (Campbell et al., 2024) and Generator Matching (Holderrieth et al., 2024) have verified the effectiveness of **multimodal models in native state spaces** in protein design. Motivated by these successes, we propose a general framework for building multimodal diffusion models on arbitrary state spaces without the need for data format unifiers. Our contributions are three-fold:

1. We propose a novel framework for building multimodal diffusion models by combining the native diffusion models designed for each data modality, and derive a unified learning objective. Under our design, learning multimodal diffusion models is as straightforward as performing a joint optimization on a sum of unimodal learning losses, despite requiring a non-trivial proof.

2. We introduce decoupled noise schedules for each data modality and theoretically justify the validity of score learning under the presence of multiple time variables. We demonstrate that this design enables us to simultaneously handle both unconditional and conditional multimodal generation in one single model. We also propose a novel guidance mechanism effective in both use cases for enhancing generation quality.

3. We experiment with text-image generation and mixed-type tabular data synthesis, achieving competitive performance on both tasks with more parameter-efficient models, without relying on pre-trained models or powerful extra encoders. More importantly, we devise a set of training strategies for the task of text-image, which is crucial for achieving success.

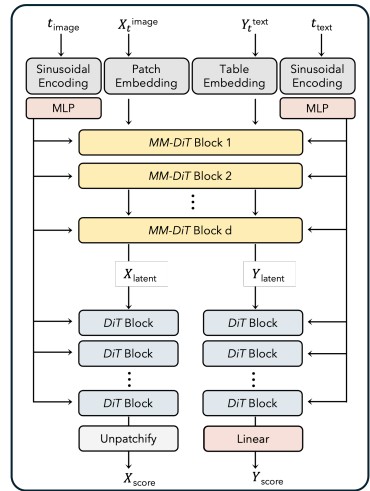 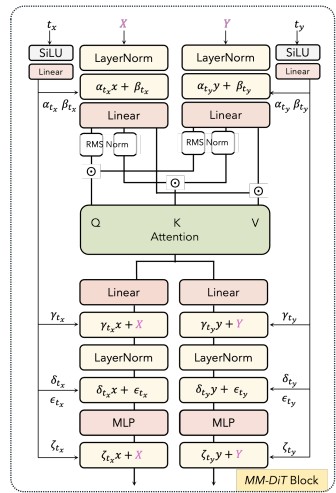 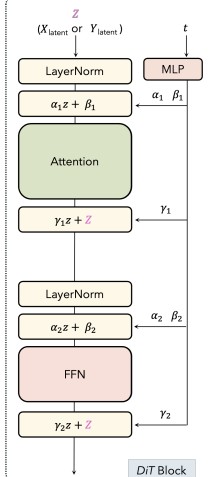

*Figure 2.* Network backbone for text-image generation, motivated by MMDiT (Esser et al., 2024) and DiT (Peebles & Xie, 2023).

## 2. Preliminaries

In this section, we review basic concepts and formulations of common diffusion models in different state spaces.

### 2.1. Continuous Space Diffusion Models

For continuous diffusion models (Song et al., 2020; Ho et al., 2020), one considers a continuous time stochastic differential equation (SDE) $\{X_t\}_{0 \leq t \leq T}$ on the Euclidean space $\mathbb{R}^d$ as the *forward process*. The process is characterized by the following dynamic,

$$\mathrm{d}X_t = \boldsymbol{f}(X_t, t)\mathrm{d}t + \boldsymbol{g}(t)\mathrm{d}W_t,$$

where $\boldsymbol{f} : \mathbb{R}^d \times \mathbb{R} \to \mathbb{R}^d$ is the drift and $\boldsymbol{g} : \mathbb{R} \to \mathbb{R}$ is the diffusion coefficient. We denote $p_t = \mathrm{Law}(X_t)$. $\boldsymbol{f}$ and $\boldsymbol{g}$ are often chosen that $p_T$ is an easy-to-sample distribution. A popular pick is the time re-parametrized Ornstein Uhlenbeck process, which corresponds to the selection of $\boldsymbol{f}(X_t, t) = -\frac{1}{2}\beta_t X_t$ and $\boldsymbol{g}(t) = \sqrt{\beta_t}$, for some positive noise schedule $\beta_t$. In such case, $p_T \approx \mathcal{N}(0, I)$ for reasonably large $T$. It can be shown that the backward process is another SDE with a different drift (Anderson, 1982),

$$\mathrm{d}X_t = \boldsymbol{f}(X_t, t)\mathrm{d}t - \boldsymbol{g}^2(t)\nabla_x \log p_t(X_t)\mathrm{d}t + \boldsymbol{g}(t)\mathrm{d}W_t$$

The common practice in training is to define the score vector $\boldsymbol{s}(X_t, t) = \nabla_x \log p_t(X_t)$ and we approximate it with a neural network $\boldsymbol{s}_\theta$, estimated by minimizing a variant of the following score matching loss (Vincent, 2011),

$$\min_\theta \int_0^T \mathbb{E}_{X_t \sim p_t}\left[\left\|\boldsymbol{s}_\theta(X_t, t) - \boldsymbol{s}(X_t, t)\right\|^2\right]\mathrm{d}t. \quad (1)$$

### 2.2. Discrete Space Diffusion Models

For discrete diffusion models (Campbell et al., 2022; Lou et al., 2024; Ou et al., 2025; Sahoo et al., 2024; Shi et al.,

2024), one considers a continuous time markov chain (CTMC) $\{X_t\}_{0 \leq t \leq T}$ on a finite state space $\mathbb{X}$ as the *forward process*. The distribution of $X_t$ is represented by a vector $p_t$ in the probability simplex on $\mathbb{R}^{|\mathbb{X}|}$. The dynamic of $X_t$ can be characterized by the following equation,

$$\frac{\mathrm{d}p_t}{\mathrm{d}t} = \boldsymbol{Q}_t p_t, \text{ where } \boldsymbol{Q}_t = (Q_t(x, y))_{x, y \in \mathbb{X}}$$

is a transition matrix satisfying that for any $x \in \mathbb{X}$, $Q_t(x, x) = -\sum_{y \neq x} Q_t(y, x)$, and for any $x \neq y \in \mathbb{X}$, $Q_t(x, y) \geq 0$. We will also denote the dynamic of $X_t$ using the following notation,

$$X_t \sim \mathrm{CTMC}(\boldsymbol{Q}_t)$$

$\boldsymbol{Q}_t$ is often chosen such that $p_T$ is a simple distribution, such as uniform on $\mathbb{X}$ or Dirac on a masked state. Common choices include uniform or masked transition matrix (Lou et al., 2024). It is known that the backward process is another process of the same form but with a different transition rate matrix (Kelly, 2011), which can be described as

$$X_t \sim \mathrm{CTMC}(\overline{\boldsymbol{Q}}_t)$$

where the rate matrix $\overline{\boldsymbol{Q}}_t = (\overline{Q}_t(x, y))_{x, y \in \mathbb{X}}$ is defined as,

$$\overline{Q}_t(y, x) = \begin{cases} \frac{p_{T-t}(y)}{p_{T-t}(x)}Q_{T-t}(x, y), & x \neq y \in \mathbb{X}, \\ -\sum_{y' \neq x} \overline{Q}_t(y', x), & x = y \in \mathbb{X}. \end{cases}$$

In discrete diffusion model training, one usually defines the concrete score vector $\boldsymbol{s}(X_t, t) = \left(\frac{p_t(y)}{p_t(X_t)}\right)_{y \in \mathbb{X}}$, and we approximate it with a neural network $\boldsymbol{s}_\theta(X_t, t)$, estimated by minimizing the following a variant of the following score

entropy loss (Benton et al., 2024; Lou et al., 2024),

$$\min_\theta \int_0^T \mathop{\mathbb{E}}_{X_t \sim p_t} \left[ \sum_{y \neq X_t} Q_t(X_t, y) \Big( \boldsymbol{s}(X_t, t)_y \log \frac{\boldsymbol{s}(X_t,t)_y}{\boldsymbol{s}_\theta(X_t,t)_y} \right.$$
$$\left. - \boldsymbol{s}(X_t,t)_y + \boldsymbol{s}_\theta(X_t,t)_y \Big) \right] \mathrm{d}t. \qquad (2)$$

### 2.3. Riemannian Diffusion Models

For Riemannian diffusion models, e.g. (De Bortoli et al., 2022; Huang et al., 2022), one considers a continuous time SDE $\{X_t\}_{0 \leq t \leq T}$ on the manifold $\mathcal{M}$ as the *forward process*, characterized by the following dynamic,

$$\mathrm{d}X_t = \boldsymbol{f}(X_t, t)\mathrm{d}t + \mathrm{d}W_t^{\mathcal{M}},$$

where $\boldsymbol{f} : \mathcal{M} \times \mathbb{R} \to T_x\mathcal{M}$ is the drift, and $\mathrm{d}W_t^{\mathcal{M}}$ is the manifold Brownian motion. For a compact manifold $\mathcal{M}$, one can pick $\boldsymbol{f} = 0$, and the corresponding $p_T \approx$ Uniform$(\mathcal{M})$ is the uniform distribution on the manifold for large $T$. It's proved that the backward process is another SDE on the manifold (De Bortoli et al., 2022),

$$\mathrm{d}X_t = \boldsymbol{f}(X_t, t)\mathrm{d}t - \nabla \log p_t(X_t)\mathrm{d}t + \mathrm{d}W_t^{\mathcal{M}},$$

where $\nabla$ is the Riemannian gradient on $\mathcal{M}$. For Riemannian diffusion model training, similar to the continuous case, one defines the score $\boldsymbol{s}(X_t, t) = \nabla \log p_t(X_t)$ and approximates it with a neural network $\boldsymbol{s}_\theta$, which typically requires special design to meet the requirement $\boldsymbol{s}_\theta(X_t, t) \in T_{X_t}\mathcal{M}$. Learning is performed through a variant of Riemannian score matching (De Bortoli et al., 2022; Huang et al., 2022),

$$\min_\theta \int_0^T \mathbb{E}_{X_t \sim p_t} \left[ \left\| \boldsymbol{s}_\theta(X_t, t) - \boldsymbol{s}(X_t, t) \right\|_{\mathcal{M}}^2 \right] \mathrm{d}t. \quad (3)$$

## 3. Methodology

In this section, we present the framework for constructing multimodal diffusion models on general state spaces in their native forms. We first discuss a unified perspective on diffusion models and then present a learning algorithm for diffusion generative modeling of multiple data modalities, where each modality has an independent time variable. Such a framework allows **any-to-any modality generation** by one single model, which simultaneously includes the **joint unconditional generation of all modalities** and the **conditional generation of a subset of modalities given the rest**. Finally, we discuss the Continuous-Discrete Multimodal Diffusion as an application of this framework to illustrate its importance and flexibility.

### 3.1. Unified Perspective on Unimodal Diffusion Models

While common unimodal diffusion models (continuous, discrete, Riemannian, etc) have distinct forward/backward processes, learning objectives, and score parameterizations,

they are essentially the realization of *denoising Markov models* (Benton et al., 2024; Ren et al., 2025b) in different situations. At a high level, diffusion models consist of a Markovian forward process that gradually injects 'noise' to transform the target data distribution into a simple distribution, and a backward generative process that inverts it using information learned from the marginals of the forward process. This abstract view is formally summarized below.

Consider a Markov process $\{X_t\}_{0 \leq t \leq T}$ with $X_0 \sim p_{\text{data}}$, defined on a state space $\mathcal{X}$. A Markov process can be conveniently characterized using the notion of infinitesimal generators. Since $X_t$ is not necessarily time-homogeneous, we instead consider the augmented process $\bar{X} = (X_t, t)$ defined on the augmented space $\mathcal{X}_* = \mathcal{X} \times [0, +\infty)$. Under mild regularity assumptions, $\bar{X}$ is a Feller process, and its generator $\mathcal{L}$ can be defined as,

$$\mathcal{L}f(x) = \lim_{t \to 0} \frac{\mathbb{E}[f(X_t)|X_0 = x] - f(x)}{t},$$

and $f : \mathcal{X}_* \to \mathbb{R}$ is a class of test function. We can understand the generator through its decomposition $\mathcal{L} = \partial_t + \hat{\mathcal{L}}$, where $\hat{\mathcal{L}}$ is an operator that acts on functions defined on the original space $\mathcal{X}$. We can also unify the characterization of the evolution of marginals $p(\cdot, t) = \text{Law}(X_t)$ as well as the score learning objectives in terms of $\mathcal{L}$. Under weak technical assumptions, $p(\cdot, t)$ satisfies the following general form of Fokker-Planck equation,

$$\partial_t p(x, t) = \hat{\mathcal{L}}^* p(x, t), \ p(x, 0) = p_{\text{data}}(x).$$

where $\hat{\mathcal{L}}^*$ is the adjoint operator of $\hat{\mathcal{L}}$. Moreover, we can define a *generalized explicit score matching* objective (Lyu, 2012; Benton et al., 2024),

$$\mathcal{J}_{\text{ESM}}(\beta) = \mathop{\mathbb{E}}_{t, p_t} \left[ \Phi\left(\frac{p}{\beta}\right)(X_t, t) \right] \qquad (4)$$

where $\Phi(f) = f^{-1}\mathcal{L}f - \mathcal{L}\log f$, and $\beta : \mathcal{X} \times [0, +\infty) \to \mathbb{R}^+$. Note that this is an extension of the common score matching objectives, and we could interpret $\mathcal{J}_{\text{ESM}}$ as a loss function that compares the 'gradient log' of $p(x, t)$ to that of $\beta(x, t)$, through which we learn the information of the forward marginal $p_t$. For example, in continuous diffusion with $\boldsymbol{g}(t) = 1$, $\mathcal{L} = \partial_t + \boldsymbol{f} \cdot \nabla + \frac{1}{2}\Delta$, we recover the explicit score matching objective on Euclidean space (Hyvärinen & Dayan, 2005), with $\Phi(p/\beta) = \frac{1}{2}\|\nabla \log p - \nabla \log \beta\|^2$. Note that the explicit score matching objectives are not tractable for training purposes; we will later introduce their equivalent, trainable variants.

### 3.2. Versatile Multimodal Diffusion Models with Decoupled Times

Building on this unified description of unimodal diffusion models, we extend the denoising Markov model framework

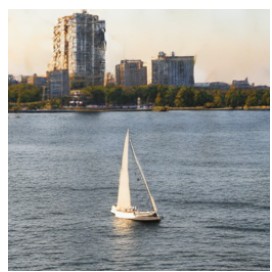

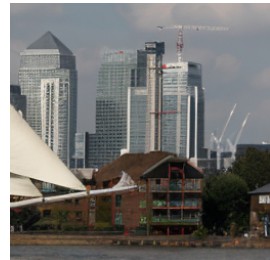

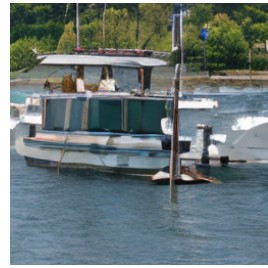

(a) **Image conditioned on text.** Sample generated using the caption: *"The image features a sailboat sailing on a large body of water, with a city skyline in the background."*

(b) **Text conditioned on Image.** We generate the following caption: *"The image features a cityscape with a large building, a bridge, and a city skyline in the background. The city is situated near the water. "*

(c) **Joint generation of text and image.** The caption corresponding to this image was: *"The image features a large white boat with a blue roof, which allows people to look out. The boat is traveling through a body of water."*

*Figure 3.* Visualization of samples generated by our approach. Captions are truncated for brevity.

to a multimodal scenario. This enables us to perform generative modeling of data distributions consisting of mixed-type data without requiring complicated preprocessing pipelines.

We begin by formally defining the *forward process* in terms of generators of Markov processes. Assume that we have a data distribution $p_{\text{data}}$ defined on the product state spaces $\mathcal{X}^1 \times \cdots \times \mathcal{X}^n$. For $1 \leq i \leq n$, we have a Markov process $\{X_{t^i}^i\}_{1 \leq t^i \leq T}$ on $\mathcal{X}^i$, which is regular enough with a unique, easy-to-sample stationary distribution $\pi^i$. We pick $T$ so that $\text{Law}(X_T^i) \approx \pi^i$ for each $1 \leq i \leq n$. Now we can introduce the following joint forward process,

$$\boldsymbol{X_t} = (X_{t^1}^1, \ldots, X_{t^i}^i, \ldots, X_{t^n}^n), 0 \leq t^1, \ldots, t^n \leq T$$
$$(X_0^1, \ldots, X_0^i, \ldots, X_0^n) \sim p_{\text{data}}(\boldsymbol{x}) \quad (5)$$

where $\boldsymbol{t} = (t^1, \ldots, t^n)$, $\boldsymbol{x} = (x^1, \ldots, x^n)$. We consider the $i$-th augmented process $\overline{X^i} = (X_{t_i}^i, t_i) \in \mathcal{X}_*^i = \mathcal{X}^i \times [0, +\infty)$ and we denote its generator as $\mathcal{L}_{X^i}$. We slightly abuse the notation and define the application of $\mathcal{L}_{X^i}$ to a multivariable test function as the following,

$$\mathcal{L}_{X^i} f(\boldsymbol{x}) = \lim_{t^i \to 0} \frac{\mathbb{E}[f(x^1, \ldots X_{t^i}^i, \ldots, x^n)|X_0^i = x^i] - f(\boldsymbol{x})}{t^i}$$

We assume that $X_{t^i}^i$ are independent Markov processes when conditioned on initial conditions. We want to emphasize that the design of this forward process (5) is not only for injecting probabilistically independent 'noises' into each modality. More importantly, it allows each modality to be noised in an **asynchronous** way.

To fully characterize and understand this forward process as a whole, we need to learn the full joint marginal of $\boldsymbol{X_t}$, which we denote as $p(\boldsymbol{x}, \boldsymbol{t})$ and should be understood as the joint distribution of $X^1, \ldots, X^n$ at time $t^1, \ldots, t^n$. To visualize this idea, we demonstrate the forward process with two independent time variables in Fig. 1.

To 'invert' this forward process for generative modeling purposes, we will need to learn information from the forward process, similar to the case of unimodal diffusion models where one performs score matching. Extending the framework of (Benton et al., 2024), we introduce the following generalized explicit score matching loss (GESM) for learning the full marginal $p(\boldsymbol{x}, \boldsymbol{t})$.

$$\mathcal{I}_{\text{GESM}} =$$
$$\mathbb{E}_{\boldsymbol{t}, \boldsymbol{x_t} \sim p(\cdot, \boldsymbol{t})} \left[ \sum_{i=0}^n \frac{\mathcal{L}_{X^i}(p/\beta_\theta)(\boldsymbol{x_t}, \boldsymbol{t})}{(p/\beta_\theta)(\boldsymbol{x_t}, \boldsymbol{t})} - \mathcal{L}_{X^i} \log(p/\beta_\theta)(\boldsymbol{x_t}, \boldsymbol{t}) \right]$$

here $\beta_\theta : \mathcal{X}_*^1 \times \cdots \times \mathcal{X}_*^n \to \mathbb{R}^+$ is our parameterized unnormalized distribution. We have the following important properties that characterize the optimizer of $\mathcal{I}_{\text{GESM}}$

**Theorem 1.** $\mathcal{I}_{\text{GESM}} \geq 0$, *with equality reached when* $\beta_\theta(\boldsymbol{x}, \boldsymbol{t}) \propto p(\boldsymbol{x}, \boldsymbol{t})$.

Thm. 1 states that the minimizer of $\mathcal{I}_{\text{GESM}}$ is $p(\boldsymbol{x}, \boldsymbol{t})$ up to a multiplicative constant. In practice, we often do not directly model $\beta_\theta$, but instead parameterize its score functions, which are invariant to multiplicative constants. For example, in continuous diffusion, one often parameterizes $\nabla \log \beta_\theta$. This makes the optimization of $\mathcal{I}_{\text{GESM}}$ a well-defined problem with a unique minimizer in terms of score learning. In practice, one can't evaluate $\mathcal{I}_{\text{GESM}}$ as it's intractable due to the true marginals $p$ being unavailable a priori. Luckily, one can efficiently compute the following denoising and implicit variants of $\mathcal{I}_{\text{GESM}}$ for learning purposes.

**Theorem 2.** $\mathcal{I}_{\text{GESM}}, \mathcal{I}_{\text{GDSM}}$ *and* $\mathcal{I}_{\text{GISM}}$ *are equivalent up to constants, where*

$$\mathcal{I}_{\text{GDSM}} =$$
$$\mathbb{E}_{\substack{\boldsymbol{t}, p_0, \\ p_{\boldsymbol{t}|0}}} \left[ \sum_{i=1}^n \frac{\mathcal{L}_{X^i}(p_{\boldsymbol{t}|0}/\beta_\theta)(\boldsymbol{x_t}, \boldsymbol{t})}{(p_{\boldsymbol{t}|0}/\beta_\theta)(\boldsymbol{x_t}, \boldsymbol{t})} - \mathcal{L}_{X^i} \log(p_{\boldsymbol{t}|0}/\beta_\theta)(\boldsymbol{x_t}, \boldsymbol{t}) \right]$$

$$\mathcal{I}_{\text{GISM}} = \mathbb{E}_{\boldsymbol{t}, p_{\boldsymbol{t}}} \left[ \sum_{i=1}^n \frac{\mathcal{L}_{X^i}^* \beta_\theta(\boldsymbol{x_t}, \boldsymbol{t})}{\beta_\theta(\boldsymbol{x_t}, \boldsymbol{t})} - \mathcal{L}_{X^i}^* \log(\beta_\theta)(\boldsymbol{x_t}, \boldsymbol{t}) \right]$$

## 3.3. Continuous-Discrete Multimodal Diffusion

We consider a distribution $p_{\text{data}}(x, y)$ where $x \in \mathbb{R}^d$, $y \in \mathbb{X}$, where $\mathbb{X}$ is a finite space. Important applications include text-image joint generation, mixed-type tabular data synthesis, etc. In this case, the natural choices on each state space would be continuous diffusion on $\mathbb{R}^d$ and discrete diffusion on $\mathbb{X}$. This results in the following forward process,

$$\begin{cases} \mathrm{d}X_t = f(X_t, t)\mathrm{d}t + g(t)\mathrm{d}B_t \\ Y_s \sim \mathrm{CTMC}(Q_s), \ (X_0, Y_0) \sim p_{\text{data}}(x, y) \end{cases} \quad (6)$$

Let's denote the joint marginal of $(X_t, Y_s)$ as $p_{t,s}(X_t, Y_s)$. with these choices of Markov processes, $\mathcal{L}_X = \partial_t + f \cdot \nabla + \frac{1}{2}g^2(t)\Delta$, and $\mathcal{L}_Y = \partial_s + Q_s$. Therefore, we can compute the generalized denoising score matching objective as,

**Proposition 1.** *For forward process* (6)*, $\mathcal{I}_{\text{GDSM}}$ is equivalent to the following objective,*

$$\mathbb{E}_{\substack{t,s,x_0,y_0 \sim p_0 \\ x_t, y_s \sim p_{t,s|0}}} \Big[ \frac{1}{2}g^2(t)\|s_\theta^X - \nabla \log p_t(x_t|x_0)\|^2 +$$

$$\sum_{y \neq y_s} Q_s(y_s, y)\Big( (s_\theta^Y)_y - \frac{p_s(y|y_0)}{p_s(y_s|y_0)} \log(s_\theta^Y)_y \Big) \Big]$$

*where $s_\theta^X, s_\theta^Y$ is the learned continuous/discrete score.*

Importantly, Prop. 1 shows an amazing result that score functions of **multimodal joint marginal** $p(\cdot, \cdot, t, s)$ can be learned through score matching with **unimodal conditional score** for each modality. This is a non-trivial result as $\mathcal{I}_{\text{GDSM}}$ suggests that, to perform multimodal generation, we need to match the score network with conditional scores of the joint distribution such as $\nabla \log p_{t,s}(x_t, y_s|x_0, y_0)$ instead of $\nabla \log p_t(x_t|x_0)$. However, thanks to Bayes' rule and the design of independent noise injection per modality, the two conditional scores are **identical**. Thus, while learning multimodal diffusion models may seem as simple as jointly optimizing a sum of unimodal diffusion model training objectives, the theoretical support for such a naive approach is grounded much deeper.

With the learned score functions, we can 'invert' the forward process for generative purposes, which is stated in Prop. 2.

**Proposition 2.** *The following process $(X_t, Y_s)$ has marginal distribution equals to $p(x, y, T - t, T - s)$,*

$$\begin{cases} \mathrm{d}X_t = -f(X_t, T - t) + g^2(T - t)\nabla \log \overleftarrow{p}_{t,s}(X_t, Y_s)\mathrm{d}t \\ \quad + g(T - t)\mathrm{d}B_t \\ Y_s \sim \mathrm{CTMC}\big(\overleftarrow{Q}(X_t, t, s)\big), \ (X_0, Y_0) \sim p(x, y, T, T) \end{cases}$$

*where $\overleftarrow{p}_{t,s} = p_{T-t,T-s}$, $\overleftarrow{Q}(X_t, t, s)$ is a rate matrix with $y', y$ entry being $\frac{\overleftarrow{p}_{t,s}(X_t, y')}{\overleftarrow{p}_{t,s}(X_t, y)}(Q_{T-s})_{yy'}$ when $y' \neq y$.*

We note that this 'backward process' is not the time reversal of the forward process in a strict sense, as it involves introducing multiple time variables into the system. However,

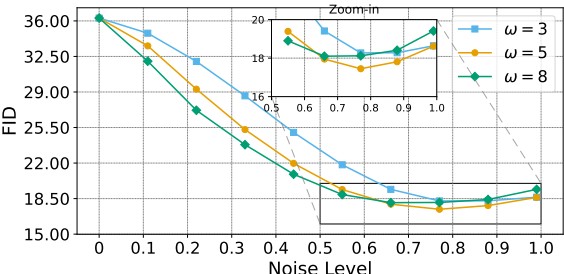

*Figure 4.* Performance of noisy guidance on MS-COCO FID-10K. We note that using partially noised conditions results in a better performance. A guidance interval of $t \in [0.3, 0.8]$ was used.

this enables us to design versatile conditional and unconditional generative sampling algorithms by choosing different value combinations of times $(t, s)$. We defer the detailed discussion of design choices and sampling algorithms for continuous-discrete multimodal diffusion to App. B.

**Multimodal unconditional generation.** To jointly generate clean data distributed as $p_{\text{data}}(x, y)$, we introduce a simulation time variable $u \in [0, T]$, and pick a time parameterization $t = \alpha_1(u)$, $s = \alpha_2(u)$ such as $\alpha_i : [0, T] \to [0, T]$ is continuous, non-decreasing with $\alpha_i(0) = 0$, $\alpha_i(T) = T$ for $i = 1, 2$. With this time-reparameterization for both time variables, we make the backward process a valid, ready-to-simulate process. This enables us to start from pure noise and generate samples from the data distribution unconditionally for both modalities. We can also choose a singular time re-parameterization so that the simulation amounts to a 2-stage approach for multimodal generation, where we first generate one modality and sample the rest conditionally based on the generated sample.

**Unimodal conditional generation.** This framework with decoupled time also enables conditional generation of modalties by simulating the associated backward process. We have the following simple but important observations,

- Given a partially noisy text $Y_s$ and its noise level $s$, simulating the $X$ backward dynamics generates a sample $X_T \sim p_{\text{data}}(x|Y_s, s)$

- Given a partially noisy image $X_t$ and its noise level $t$, simulating the $Y$ backward dynamics generates a sample $Y_T \sim p_{\text{data}}(y|X_t, t)$.

Note that the choices of $t$ or $s$ in the conditioning are not restricted. When picking $t$ or $s$ as $T$, this is equivalent to single-modality unconditional generation, as $X_T$ or $Y_T$ are pure noise. When picking $t$ or $s$ as $0$, this is equivalent to conditional generation, as $X_0$ or $Y_0$ are clean data samples. More interestingly, when picking $0 < s, t < T$ as conditions, we generate samples based on partially noised conditions. This gives rise to the following new guidance mechanism for enhancing generation quality.

*Table 1.* Results on the **text to image conditional generation** on MS-COCO. We mark the extra encoders leveraged by each model with the corresponding sizes and types. SR: super resolution, TE: text encoder, VAE: variational autoencoder, VE: visual encoder, VQ-GAN: Vector Quantized GAN, VQ-VAE: vector-quantized variational autoencoder.

| Model | FID | Number of Images | #Params | Extra Encoders |
|---|---|---|---|---|
| Models for Text-to-Image generation only | | | | |
| DALL-E 2 (Ramesh et al., 2022) | 10.39 | 650M | 6.5B | 123M (TE) + 700M (SR) |
| Imagen (Saharia et al., 2022) | 7.27 | 860M | 3B | 4.6B (TE) + 600M (SR) |
| Stable Diffusion (Rombach et al., 2022) | 12.63 | 400M | 1.45B | 123M (TE) + 83M (VAE) |
| PixArt-$\alpha$ XL/2 (Chen et al., 2023) | 7.32 | 25M | 600M | 123M (TE) + 83M (VAE) |
| MMDiT-improved (Ifriqi et al., 2024) | 6.79 | 12M | 600M | 123M (TE) + 83M (VAE) |
| Models for multimodal generation and understanding | | | | |
| Show-o (Xie et al., 2024) | 9.24 | 35M | 1.3B | 115M (VE) + 307M (VQ-VAE) |
| Transfusion (Zhou et al., 2024) | 6.78 | 692M | 7B | 86M (VAE) |
| Chameleon (Meta, 2024) | 26.7 | 600M | 7B | 307M (VQ-GAN) |
| JetFormer (Tschannen et al., 2024) | 20.86 | 1B | 2.75B | — |
| Models for multimodal generation only | | | | |
| Versatile Diffusion (Xu et al., 2023) | 11.10 | 400M | 1.45B | 123M (TE) + 83M (VAE) + 110M (TE) |
| UniD3 (Hu et al., 2022) | 25.11 | 592K | 600M | 123M (TE) + 307M (VQ-GAN) |
| Our model | 16.16 | 12M | 481M | 83M (VAE) |

## 3.4. Noisy Guidance

Guidance techniques have been a core component in modern diffusion models for improving generation quality (e.g., Dhariwal & Nichol, 2021; Ho & Salimans, 2021; Kynkäänniemi et al., 2024; Li et al., 2024b). For continuous diffusion models, with strength $\omega$, the classifier-free guidance is obtained by interpolating the unconditional and conditional score functions,

$$\omega s_\theta(x_t, t, c) + (1-\omega)s_\theta(x_t, t, \emptyset) \tag{7}$$

One perspective to understand the effectiveness of guidance methods is to view the unconditional score function as a conditional model with a fully-noised condition input, and the interpolation effectively serves as a correction of the conditional scores. However, the unconditional score might not be the best choice of correctors, as it causes an excessive trade-off between fidelity and diversity, resulting in a significant loss in the latter (Karras et al., 2024a). This raises an interesting question about whether CFG can be improved by finding a better alternative to the unconditional score in (7). In fact, within our framework, we notice that

$$s_\theta^X(x_t, y_s, t, s) \approx \nabla_x \log p_{t,s}(x_t, y_s) = \nabla_x \log p(x_t, t | y_s, s)$$

The last equality results from Bayes' theorem, and it shows that our model in fact learns conditional scores **at all noise levels**. Leveraging this fact, we propose a new form of guidance named **noisy guidance**, where the unconditional score in (7) is replaced with a class of conditional models with conditions noised to different levels:

$$\omega s_\theta(x_t, y_s, t, s) + (1-\omega)s_\theta(x_t, y_\sigma, t, \sigma) \quad \sigma > s \tag{8}$$

We note that the noisy guidance framework is a more general one, as it recovers the vanilla setting of CFG when

$s = 0$ and $\sigma = T$. The scenario of conditional generation corresponds to $s = 0$ (a clean condition $y_0$ is given), and we choose $T > \sigma > 0$ to improve generation quality with a **partially conditioned** guiding model. More interestingly, noisy guidance can even be applied to **unconditional generation** in an unsupervised way where $s > 0$. In this case, while $s$ is changing (since we are also generating $y_s$), we can still apply guidance in this process by adaptively picking $\sigma$ as long as $\sigma > s$. These observations indicate the power and robustness of noisy guidance as a by-product of our proposed multimodal diffusion model learning framework.

To showcase the performance of noisy guidance, we evaluate FID-10K on MS-COCO (Lin et al., 2014) using different values of $\sigma \in [0, 1]$. We find that partially denoising the caption results in an improved FID, as shown in Fig. 4. The superior performance of noisy guidance is possibly due to using a **partially corrupted version** of the conditional model as the guiding model. This aligns closely with the idea of Autoguidance proposed in Karras et al. (2024a), which utilizes an under-trained, smaller model as the guiding model instead of the unconditional ones. Similar to Autoguidance, noisy guidance seeks to identify and reduce the errors made by the conditional score model by measuring its difference to the partially conditioned one, boosting the generation performance. Finally, we remark that (8) is only a special case applied to continuous diffusion, and a similar idea can be adapted to other modalities, such as discrete diffusion guidance (Nisonoff et al., 2024; Schiff et al., 2024).

## 4. Experiments

To demonstrate the effectiveness and relevance of our framework for training multimodal diffusion models, we consider two tasks: text-image generation and mixed-type tab-

*Table 2.* Performance on the **Trend** metric in percentage (%). Higher values indicate better performance. Best performance in **bold**. Second best in underline.

| Methods | #Parameters | Adult | Default | Shoppers | Magic | Beijing | News |
|---|---|---|---|---|---|---|---|
| GOGGLE (Liu et al., 2023) | $\sim 5.6$M | 54.71 | 78.06 | 76.10 | 90.53 | 54.06 | 76.81 |
| STaSy (Kim et al., 2022) | $\sim 10.3$M | $85.49_{\pm 0.25}$ | $94.04_{\pm 0.26}$ | $91.51_{\pm 0.15}$ | $93.39_{\pm 0.53}$ | $92.00_{\pm 0.10}$ | $96.93_{\pm 0.04}$ |
| CoDi (Lee et al., 2023) | $\sim 25.0$M | $77.51_{\pm 0.08}$ | $31.59_{\pm 0.05}$ | $82.22_{\pm 0.11}$ | $93.47_{\pm 0.25}$ | $92.93_{\pm 0.15}$ | $88.90_{\pm 0.01}$ |
| TabDDPM (Kotelnikov et al., 2023) | $\sim 11.7$M | $96.99_{\pm 0.25}$ | $95.11_{\pm 0.10}$ | $93.39_{\pm 0.16}$ | $98.30_{\pm 0.22}$ | $97.20_{\pm 0.09}$ | $86.84_{\pm 0.11}$ |
| TABSYN (Zhang et al., 2023) | $\sim 10.7$M | $\underline{98.46}_{\pm 0.27}$ | $\mathbf{97.95}_{\pm 0.12}$ | $\underline{97.93}_{\pm 0.21}$ | $98.94_{\pm 0.31}$ | $\mathbf{97.76}_{\pm 0.28}$ | $\underline{98.56}_{\pm 0.03}$ |
| TABSYN (reproduced) | $\sim 10.7$M | $98.29_{\pm 0.22}$ | $95.25_{\pm 0.51}$ | $97.82_{\pm 0.14}$ | $\mathbf{99.16}_{\pm 0.16}$ | $94.86_{\pm 0.34}$ | $98.52_{\pm 0.09}$ |
| Our model | $\sim \mathbf{64}$K | $\mathbf{98.75}_{\pm 0.09}$ | $\underline{96.00}_{\pm 1.23}$ | $\mathbf{98.24}_{\pm 0.13}$ | $98.85_{\pm 0.42}$ | $\underline{97.42}_{\pm 0.11}$ | $\mathbf{98.57}_{\pm 0.16}$ |

*Table 3.* Performance on the **MLE** metric. Higher values in AUC and lower values in RMSE indicate better testing performance. Best performance in **bold**. Second best in underline.

| Methods | #Parameters | Adult (AUC↑) | Default (AUC↑) | Shoppers (AUC↑) | Magic (AUC↑) | Beijing (RMSE↓) | News (RMSE↓) |
|---|---|---|---|---|---|---|---|
| GOGGLE (Liu et al., 2023) | $\sim 5.6$M | $.778_{\pm 0.012}$ | $.584_{\pm 0.005}$ | $.658_{\pm 0.052}$ | $.654_{\pm 0.024}$ | $1.090_{\pm 0.025}$ | $.877_{\pm 0.002}$ |
| STaSy (Kim et al., 2022) | $\sim 10.3$M | $.906_{\pm 0.001}$ | $.752_{\pm 0.006}$ | $.914_{\pm 0.005}$ | $.934_{\pm 0.003}$ | $.656_{\pm 0.014}$ | $.871_{\pm 0.002}$ |
| CoDi (Lee et al., 2023) | $\sim 25.0$M | $.871_{\pm 0.006}$ | $.525_{\pm 0.006}$ | $.865_{\pm 0.006}$ | $.932_{\pm 0.003}$ | $.818_{\pm 0.021}$ | $1.21_{\pm 0.005}$ |
| TabDDPM (Kotelnikov et al., 2023) | $\sim 11.7$M | $.907_{\pm 0.001}$ | $.758_{\pm 0.004}$ | $.918_{\pm 0.005}$ | $.935_{\pm 0.003}$ | $.592_{\pm 0.011}$ | $4.86_{\pm 3.04}$ |
| TABSYN (Zhang et al., 2023) | $\sim 10.7$M | $\mathbf{.915}_{\pm 0.002}$ | $\mathbf{.764}_{\pm 0.004}$ | $\underline{.920}_{\pm 0.005}$ | $.938_{\pm 0.002}$ | $\underline{.582}_{\pm 0.008}$ | $\underline{.861}_{\pm 0.027}$ |
| TABSYN (reproduced) | $\sim 10.7$M | $.910_{\pm 0.001}$ | $.755_{\pm 0.004}$ | $.916_{\pm 0.004}$ | $\underline{.939}_{\pm 0.003}$ | $.655_{\pm 0.012}$ | $\mathbf{.851}_{\pm 0.024}$ |
| Our model | $\sim \mathbf{64}$K | $\mathbf{.915}_{\pm 0.001}$ | $\mathbf{.764}_{\pm 0.002}$ | $\mathbf{.924}_{\pm 0.003}$ | $\mathbf{.941}_{\pm 0.002}$ | $\mathbf{.543}_{\pm 0.012}$ | $.864_{\pm 0.021}$ |

ular data synthesis, both of which are accomplished using the continuous-discrete Multimodal diffusion discussed in Sec. 3.3. The results for text-image generation are presented in Sec. 4.1 and the results for tabular data synthesis are presented in Sec. 4.2. e also include results for combining Riemannian and discrete diffusion in App. E to demonstrate the generality of the framework.

### 4.1. Text-Image Generation

**Architecture** We design a new score network backbone for this task based on the celebrated success of Diffusion Transformer (DiT) (Peebles & Xie, 2023) and Multimodal Diffusion Transformer (MMDiT) (Esser et al., 2024). We first process the inputted (noisy) images and texts by passing them through an MMDiT with a per-modality unique time conditioning. MMDiT's remarkable strength in modeling cross-modal interaction, as well as allowing independent conditioning for each modality, makes it ideal for the backbone. The tokens then undergo unimodal DiTs for a more refined learning process. We present a comprehensive diagram of the backbone in Fig. 2 and refer the reader to App. C for further details.

**Datasets** We train on the SAM-LLaVA dataset introduced by Chen et al. (2023). This dataset is constructed by adding captions to the Segment Anything (SAM) (Kirillov et al., 2023) using LLaVA (Liu et al., 2024), which results in rich and diverse captions. However, it suffers from hallucinations of LLava. For example, many colored images are described as being black and white. Following (Chen et al., 2023), we tokenize each caption with a length of 120 tokens.

**Training & Evaluation** We train our model using a multi-stage training strategy. We kindly refer readers to App. C for more details. We evaluate FID-30K on MS-COCO (Lin et al., 2014). Compared with SAM-LLaVA, MS-COCO comes with much shorter captions. To address this distribution shift in caption length between training and inference time, we draw inspiration from (Ifriqi et al., 2024) and replicate the text to increase the caption size. We also limit the number of tokens to 40 during this evaluation. We compare our results to other methods in Tab. 1. In terms of text-to-image (T2I), our methods produce similar performance compared to many other industrial-level models with larger model sizes. Notably, our model is trained on fewer samples, features a significantly smaller backbone, and does not utilize extra foundation models to aid multimodal representation learning. This reflects both the efficiency and the effectiveness of our proposed approach. We also present qualitative samples in Fig. 3. For evaluation of image-to-text (I2T) and joint generation, please kindly see App. F.

### 4.2. Mixed-type Tabular Data Synthesis

**Architecture** We devise a score network based on DiT for this task, where both discrete and continuous tabular data are fed into the transformer after simple dimension rescaling. Our design aims at achieving early fusion of both modalities for more efficient learning. See App. D.3 for details.

**Datasets** We experiment on 6 real-world tabular datasets acquired from UCI Machine Learning Repository[1]. Every dataset contains columns of numerical or categorical features, associated with binary classification tasks or regres-

---
[1]https://archive.ics.uci.edu/

sion tasks. Detailed descriptions of datasets are in App. D.2.

**Training & Evaluation** We describe the score model architecture and training settings in App. D.3. For evaluation, we follow the same setting as Zhang et al. (2023). We evaluate the Trend, Machine Learning Efficiency (MLE), Shape, Precision, and Recall of the generated data. We present results on **Trend** in Tab. 2 and **MLE** in Tab. 3. For more results on other metrics, please see App. D.4. All experiments are repeated 20 times for robustness.

As evident from Tab. 2 and Tab. 3, our model performs either the best or second-best, with a negligible gap, among most datasets. Notably, our model has only $\sim 64$K parameters, which is **100 to 200 times smaller** than models used in other methods. The significant reduction in model size stems from the fact that our method operates natively on the state space of mixed-type tabular data, eliminating parameter-heavy encoders like VAE. Our newly designed transformer-based score network leverages the concept of early fusion (Meta, 2024), which also enhances parameter efficiency. This network learns a joint embedding between modalities starting from the first attention layer and is more parameter-efficient. These results again showcase the efficiency and advantage of our proposed multimodal diffusion model framework.

## 5. More Information on Related Works

**Multimodal Generative Models.** Various works in the literature have attempted joint generation of multimodal data. Many existing methods approach the task by leveraging autoregressive models by first tokenizing multimodal data into discrete tokens and then generating them sequentially (Meta, 2024; Xie et al., 2024; Zhou et al., 2024; Tschannen et al., 2024; Lu et al., 2024; Ge et al., 2024; Wu et al., 2024; Wang et al., 2024a). Another portion of the algorithms is built on the versatile capability of diffusion/flow-based methods to generate latent representations of multimodal data (Lee et al., 2023; Bao et al., 2023b; Ruan et al., 2023; Hu et al., 2022; Zhang et al., 2023; Kim et al., 2024; Chen et al., 2024a; Li et al., 2024a; Swerdlow et al., 2025; Hayes et al., 2025; Bai et al., 2025; Shi et al., 2025). One thing in common among the aforementioned approaches is that they all extensively utilize tokenizers or encoders to produce a unimodal latent space for multimodal data, which are not modular and heavily tailored to specific applications with little theoretical support. In contrast to these works in the literature, we propose a general multimodal diffusion learning framework in this paper, which is flexible for generating data on arbitrary state spaces. This is achieved by minimizing the need for external, modality-specific tokenizers and encoders, while keeping the generation in the native spaces of the targeted data.

**Decoupled Time Variables.** It's worth noting that the decoupled time design, essential to our proposed framework, has in fact been explored by many application-driven works in the literature, such as UniDiffuser (Bao et al., 2023a), MultiFlow (Campbell et al., 2024), AVDiT (Kim et al., 2024), and OmniFlow (Li et al., 2024a). While these methods all leverage the multiple time variable design to achieve any-to-any generation, they mostly consider this design as a trick and do not investigate it from an algorithmic perspective. Our work contributes to this literature by deriving the unified training objective and backward generative process in the presence of multiple time variables, providing theoretical justification for the validity of this design. To the best of our knowledge, our work is the first to formalize this idea and generalize it to an arbitrary number of modalities.

A highly relevant work worth discussing in more detail is UniDiffuser (Bao et al., 2023b), which addresses the same text-image joint generation task considered in this paper while also utilizing decoupled time variables. A fundamental difference in algorithm design is that UniDiffuser purely relies on continuous diffusion models in the CLIP latent space, which is shared by both text and images, whereas our proposed continuous-discrete diffusion operates natively on the product space of Euclidean and finite-state spaces.

**Diffusion Models in General State Spaces.** Theoretical results have demonstrated that diffusion models can be generalized to denoising Markov models, a class of generative models constructed based on the notion of Markov processes (Benton et al., 2024; Ren et al., 2025b). These works considered unimodal diffusions on general state spaces with a single time variable. Our work extends the framework of denoising Markov models by incorporating multimodal diffusion models on the product of different state spaces, as well as multiple time variables. In the literature, there are also MultiFlow (Campbell et al., 2024) and Generator Matching (Holderrieth et al., 2024), which are multimodal extensions of flow-based methods in native state spaces. Their algorithm construction specifically focused on the task of protein sequence-structure co-generation. In contrast, we present a general recipe for multimodal diffusion models that does not initially target specific tasks, despite empirically validating the proposed framework on two examples: text-image generation and mix-type tabular data synthesis, leveraging our newly proposed Continuous-Discrete Diffusion.

## 6. Conclusions, Limitations and Future Works

We propose a novel framework for constructing multimodal diffusion models on general state spaces. We experiment on text-image and tabular data generation, and our approach achieves competitive performances with a significantly smaller model size. One limitation of this work is that we didn't explore the possibility of utilizing pretrained unimodal diffusion models as initialization of multimodal diffusion training, which could further boost training efficiency. We leave this as a future direction for investigation.

## Acknowledgements

KR, YZ and MT are grateful for partial supports by NSF Grant DMS-1847802, Cullen-Peck Scholarship, and Emory-GT AI.Humanity Award. FY is grateful for partial support by UAlbany-IBM research funding. This research was supported in part by the AI Computing Cluster at the University at Albany.

## Impact Statement

This paper presents work aimed at advancing the field of Machine Learning. There are many potential societal consequences of our work, none of which we feel must be specifically highlighted here.

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

# A. Proofs

## A.1. Proof of Theorem 1

The following proof follows a similar idea as is presented Benton et al. (2024):

*Proof.* By Jensen's inequality, we have:

$$\log\left(\mathbb{E}[f(x^1,\ldots,X_{t^i}^i,\ldots,x^n)\mid X_0^i=x^i]\right)\leq\mathbb{E}[\log(f)(x^1,\ldots,X_{t^i}^i,\ldots,x^n)\mid X_0^i=x^i]$$

This implies the following inequality:

$$\underbrace{\frac{\log\left(\mathbb{E}[f(x^1,\ldots,X_{t^i}^i,\ldots,x^n)\mid X_0^i=x^i]\right)-\log f(\boldsymbol{x})}{t^i}}_{\text{LHS}}\leq\underbrace{\frac{\mathbb{E}[\log(f)(x^1,\ldots,X_{t^i}^i,\ldots x^n)\mid X_0^i=x^i]-\log f(\boldsymbol{x})}{t^i}}_{\text{RHS}}$$

where we denote $\boldsymbol{x}=(x^1,\ldots,x^n)$. Taking the limit as $t^i\to 0$, we notice that the limit of RHS equals $\mathcal{L}_{X^i}(\log f)(\boldsymbol{x})$. On the other hand, if we consider the following function $g$:

$$g(h)=\mathbb{E}[f(x^1,\ldots,X_h^i,\ldots,x^n)\mid X_0^i=x^i],\quad g(0)=f(\boldsymbol{x})$$

We can calculate the limit of LHS as $t^i\to 0$,

$$\lim_{t^i\to 0}\text{LHS}=\lim_{t^i\to 0}\frac{\log(g(t^i))-\log g(0))}{t^i}=(\log(g))'(0)=\frac{1}{g(0)}g'(0)=\frac{\mathcal{L}_{X^i}f(\boldsymbol{x})}{f(\boldsymbol{x})}$$

Combining them, we have that

$$\frac{\mathcal{L}_{X^i}f(\boldsymbol{x})}{f(\boldsymbol{x})}\geq\mathcal{L}(\log f)(\boldsymbol{x})$$

Applying this result by choosing $f=p/\beta_\theta$, we have that

$$\frac{\mathcal{L}_{X^i}(p/\beta_\theta)(\boldsymbol{x_t},\boldsymbol{t})}{(p/\beta_\theta)(\boldsymbol{x_t},\boldsymbol{t})}-\mathcal{L}_{X^i}\log(p/\beta_\theta)(\boldsymbol{x_t},\boldsymbol{t})\geq 0 \tag{9}$$

This finishes the proof of $\mathcal{I}_{\text{GESM}}\geq 0$. To see that $\mathcal{I}_{\text{GESM}}$ is minimized when $\beta_\theta(\boldsymbol{x},\boldsymbol{t})\propto p(\boldsymbol{x},\boldsymbol{t})$, note that Jensen's inequality holds the equality sign when the test function is constant. Therefore, (9) holds the equality sign for each $\mathcal{L}_{X^i}$ whenever $p/\beta_\theta$ is identically constant, therefore $\mathcal{I}_{\text{GESM}}$ is optimized when $\beta_\theta$ is equivalent to $p$ up to a multiplicative constant. $\square$

## A.2. Proof of Theorem 2

*Proof.* We will start by showing the equivalence between $\mathcal{I}_{\text{GDSM}}$ and $\mathcal{I}_{\text{GISM}}$, and then we will demonstrate the equivalence between $\mathcal{I}_{\text{GISM}}$ and $\mathcal{I}_{\text{GESM}}$ to finish the proof. We start with the definition of $\mathcal{I}_{\text{GDSM}}$,

$$\begin{aligned}\mathcal{I}_{\text{GDSM}}&=\mathop{\mathbb{E}}_{\boldsymbol{t},p_0,p_{\boldsymbol{t}|0}}\left[\sum_{i=1}^n\frac{\mathcal{L}_{X^i}(p_{\boldsymbol{t}|0}/\beta_\theta)(\boldsymbol{x_t},\boldsymbol{t})}{(p_{\boldsymbol{t}|0}/\beta_\theta)(\boldsymbol{x_t},\boldsymbol{t})}-\mathcal{L}_{X^i}\log(p_{\boldsymbol{t}|0}/\beta_\theta)(\boldsymbol{x_t},\boldsymbol{t})\right]\\&=\mathop{\mathbb{E}}_{\boldsymbol{t},p_0}\int_{\mathcal{X}}p_{\boldsymbol{t}|0}(\boldsymbol{x_t},\boldsymbol{t})\left[\sum_{i=1}^n\frac{\mathcal{L}_{X^i}(p_{\boldsymbol{t}|0}/\beta_\theta)(\boldsymbol{x_t},\boldsymbol{t})}{(p_{\boldsymbol{t}|0}/\beta_\theta)(\boldsymbol{x_t},\boldsymbol{t})}-\mathcal{L}_{X^i}\log(p_{\boldsymbol{t}|0}/\beta_\theta)(\boldsymbol{x_t},\boldsymbol{t})\right]\mathrm{d}\boldsymbol{x_t}\\&=\mathop{\mathbb{E}}_{\boldsymbol{t},p_0}\int_{\mathcal{X}}\left[\sum_{i=1}^n\beta_\theta(\boldsymbol{x_t},\boldsymbol{t})\mathcal{L}_{X^i}(p_{\boldsymbol{t}|0}/\beta_\theta)(\boldsymbol{x_t},\boldsymbol{t})-p_{\boldsymbol{t}|0}(\boldsymbol{x_t},\boldsymbol{t})\mathcal{L}_{X^i}\log(p_{\boldsymbol{t}|0}/\beta_\theta)(\boldsymbol{x_t},\boldsymbol{t})\right]\mathrm{d}\boldsymbol{x_t}\end{aligned}$$

Then, using the properties of the adjoint $\mathcal{L}^*_{X^i}$, we can continue to simplify the objective as,

$$\mathcal{I}_{\mathrm{GDSM}} = \underset{t,p_0}{\mathbb{E}} \int_{\mathcal{X}} \left[ \sum_{i=1}^n \mathcal{L}^*_{X^i}\beta_\theta(\boldsymbol{x_t},\boldsymbol{t}) \cdot p_{t|0}(\boldsymbol{x_t},\boldsymbol{t})/\beta_\theta(\boldsymbol{x},\boldsymbol{t}) - p_{t|0}(\boldsymbol{x_t},\boldsymbol{t})\mathcal{L}_{X^i}\log(p_{t|0}/\beta_\theta)(\boldsymbol{x_t},\boldsymbol{t}) \right] \mathrm{d}\boldsymbol{x_t}$$

$$= \underset{t,p_0}{\mathbb{E}} \int_{\mathcal{X}} p_{t|0}(\boldsymbol{x_t},\boldsymbol{t}) \left[ \sum_{i=1}^n \frac{\mathcal{L}^*_{X^i}\beta_\theta(\boldsymbol{x_t},\boldsymbol{t})}{\beta(\boldsymbol{x_t},\boldsymbol{t})} + \mathcal{L}_{X^i}\log\beta_\theta(\boldsymbol{x_t},\boldsymbol{t}) - \mathcal{L}_{X^i}\log p_{t|0}(\boldsymbol{x_t},\boldsymbol{t}) \right] \mathrm{d}\boldsymbol{x_t}$$

$$= \underset{t,p_0}{\mathbb{E}}\ \underset{p_{t|0}}{\mathbb{E}} \left[ \sum_{i=1}^n \frac{\mathcal{L}^*_{X^i}\beta_\theta(\boldsymbol{x_t},\boldsymbol{t})}{\beta_\theta(\boldsymbol{x_t},\boldsymbol{t})} + \mathcal{L}_{X^i}\log\beta_\theta(\boldsymbol{x_t},\boldsymbol{t}) \right] + \mathrm{const}$$

$$= \underset{t,p_t}{\mathbb{E}} \left[ \sum_{i=1}^n \frac{\mathcal{L}^*_{X^i}\beta_\theta(\boldsymbol{x_t},\boldsymbol{t})}{\beta_\theta(\boldsymbol{x_t},\boldsymbol{t})} + \mathcal{L}_{X^i}\log\beta_\theta(\boldsymbol{x_t},\boldsymbol{t}) \right] + \mathrm{const}$$

$$= \mathcal{I}_{\mathrm{GISM}} + \mathrm{const}$$

This finishes the proof of equivalence between $\mathcal{I}_{\mathrm{GDSM}}$ and $\mathcal{I}_{\mathrm{GISM}}$. To show the equivalence between $\mathcal{I}_{\mathrm{GISM}}$ and $\mathcal{I}_{\mathrm{GESM}}$, we start with the definition of $\mathcal{I}_{\mathrm{GESM}}$,

$$\mathcal{I}_{\mathrm{GESM}} = \underset{t,\boldsymbol{x_t}\sim p(\cdot,t)}{\mathbb{E}} \left[ \sum_{i=0}^n \frac{\mathcal{L}_{X^i}(p/\beta_\theta)(\boldsymbol{x_t},\boldsymbol{t})}{(p/\beta_\theta)(\boldsymbol{x_t},\boldsymbol{t})} - \mathcal{L}_{X^i}\log(p/\beta_\theta)(\boldsymbol{x_t},\boldsymbol{t}) \right]$$

$$= \underset{t}{\mathbb{E}} \int_{\mathcal{X}} p(\boldsymbol{x_t},\boldsymbol{t}) \left[ \sum_{i=0}^n \frac{\mathcal{L}_{X^i}(p/\beta_\theta)(\boldsymbol{x_t},\boldsymbol{t})}{(p/\beta_\theta)(\boldsymbol{x_t},\boldsymbol{t})} - \mathcal{L}_{X^i}\log(p/\beta_\theta)(\boldsymbol{x_t},\boldsymbol{t}) \right] \mathrm{d}\boldsymbol{x_t}$$

$$= \underset{t}{\mathbb{E}} \int_{\mathcal{X}} \left[ \sum_{i=0}^n \beta_\theta(\boldsymbol{x_t},\boldsymbol{t})\mathcal{L}_{X^i}(p/\beta_\theta)(\boldsymbol{x_t},\boldsymbol{t}) - p(\boldsymbol{x_t},\boldsymbol{t})\mathcal{L}_{X^i}\log(p/\beta_\theta)(\boldsymbol{x_t},\boldsymbol{t}) \right] \mathrm{d}\boldsymbol{x_t}$$

Using again the properties of ajoint $\mathcal{L}^*_{X^i}$, we have

$$\mathcal{I}_{\mathrm{GESM}} = \underset{t}{\mathbb{E}} \int_{\mathcal{X}} \left[ \sum_{i=0}^n \mathcal{L}^*_{X^i}\beta_\theta(\boldsymbol{x_t},\boldsymbol{t})(p/\beta_\theta)(\boldsymbol{x_t},\boldsymbol{t}) - p(\boldsymbol{x_t},\boldsymbol{t})\mathcal{L}_{X^i}\log(p/\beta_\theta)(\boldsymbol{x_t},\boldsymbol{t}) \right] \mathrm{d}\boldsymbol{x_t}$$

$$= \underset{t}{\mathbb{E}} \int_{\mathcal{X}} p(\boldsymbol{x_t},\boldsymbol{t}) \left[ \sum_{i=0}^n \frac{\mathcal{L}^*_{X^i}\beta_\theta(\boldsymbol{x_t},\boldsymbol{t})}{\beta_\theta(\boldsymbol{x_t},\boldsymbol{t})} - \mathcal{L}_{X^i}\log(p/\beta_\theta)(\boldsymbol{x_t},\boldsymbol{t}) \right] \mathrm{d}\boldsymbol{x_t}$$

$$= \underset{t,\boldsymbol{x_t}\sim p_t}{\mathbb{E}} \left[ \sum_{i=0}^n \frac{\mathcal{L}^*_{X^i}\beta_\theta(\boldsymbol{x_t},\boldsymbol{t})}{\beta_\theta(\boldsymbol{x_t},\boldsymbol{t})} - \mathcal{L}_{X^i}\log p(\boldsymbol{x_t},\boldsymbol{t}) + \mathcal{L}_{X^i}\log\beta_\theta(\boldsymbol{x_t},\boldsymbol{t}) \right]$$

$$= \underset{t,\boldsymbol{x_t}\sim p_t}{\mathbb{E}} \left[ \sum_{i=0}^n \frac{\mathcal{L}^*_{X^i}\beta_\theta(\boldsymbol{x_t},\boldsymbol{t})}{\beta_\theta(\boldsymbol{x_t},\boldsymbol{t})} + \mathcal{L}_{X^i}\log\beta_\theta(\boldsymbol{x_t},\boldsymbol{t}) \right] + \mathrm{const}$$

$$= \mathcal{I}_{\mathrm{GISM}} + \mathrm{const}$$

This finishes the proof of equivalence of $\mathcal{I}_{\mathrm{GDSM}}, \mathcal{I}_{\mathrm{GISM}}, \mathcal{I}_{\mathrm{GESM}}$ up to an additive, $\theta$-independent constant. $\square$

### A.3. Proof of Proposition 1

*Proof.* In the following, we derive $\mathcal{I}_{\mathrm{GDSM}}$ when choosing the forward process as in (6). Recall that $\mathcal{I}_{\mathrm{GDSM}}$ is given as,

$$\mathcal{I}_{\mathrm{GDSM}} = \underset{t,p_0,p_{t|0}}{\mathbb{E}} \left[ \underbrace{\frac{\mathcal{L}_X(p_{t|0}/\beta_\theta)(\boldsymbol{x_t},\boldsymbol{t})}{(p_{t|0}/\beta_\theta)(\boldsymbol{x_t},\boldsymbol{t})} - \mathcal{L}_X\log(p_{t|0}/\beta_\theta)(\boldsymbol{x_t},\boldsymbol{t})}_{\mathcal{J}_X} + \underbrace{\frac{\mathcal{L}_Y(p_{t|0}/\beta_\theta)(\boldsymbol{x_t},\boldsymbol{t})}{(p_{t|0}/\beta_\theta)(\boldsymbol{x_t},\boldsymbol{t})} - \mathcal{L}_Y\log(p_{t|0}/\beta_\theta)(\boldsymbol{x_t},\boldsymbol{t})}_{\mathcal{J}_Y} \right]$$

where $\boldsymbol{t} = (t,s)$, $\boldsymbol{x_t} = (X_t, Y_s)$, $p_{t|0}(\boldsymbol{x_t},\boldsymbol{t}) = p(X_t, Y_s \mid X_0 = x_0, Y_0 = y_0)$, $\mathcal{L}_X$ and $\mathcal{L}_Y$ are generator of the following dynamics.

$$\begin{cases} \mathrm{d}X_t = f(X_t, t)\mathrm{d}t + g(t)\mathrm{d}B_t \\ Y_s \sim \mathrm{CTMC}(Q_s), \ (X_0, Y_0) \sim p_{\mathrm{data}}(x,y) \end{cases}$$

See Lem. 1 for a detailed proof. We start by computing the score matching operators $\Phi_X$ and $\Phi_Y$ for $X$ and $Y$ respectively. Note that $\mathcal{L}_X$ and $\mathcal{L}_Y$ are given by the following expressions for a test function $h = h(x_t, y_s, t, s)$,

$$\mathcal{L}_X h = f \cdot \nabla h + \frac{1}{2} g(t)^2 \Delta h, \quad \mathcal{L}_Y h = \sum_{y \in \mathbb{X}} h(x_t, y, t, s) Q_s(y, y_s)$$

Therefore, for the score matching operator associated with $\mathcal{L}_X$, it can be expressed as

$$
\begin{aligned}
\Phi_X(h) &= \frac{\mathcal{L}_X h}{h} - \mathcal{L}_X \log h \\
&= \frac{f \cdot \nabla h}{h} + \frac{1}{2} g(t)^2 \frac{\Delta h}{h} - f \cdot \nabla \log h - \frac{1}{2} \Delta \log h \\
&= \frac{1}{2} g(t)^2 \cdot \frac{\nabla \cdot (h \nabla \log h)}{h} - \frac{1}{2} g(t)^2 \Delta \log h \\
&= \frac{1}{2} g(t)^2 \nabla \log h \cdot \frac{\nabla h}{h} + \frac{1}{2} g(t)^2 \nabla \cdot \nabla \log h - \frac{1}{2} g(t)^2 \Delta \log h
\end{aligned}
$$

Note that $\frac{\nabla h}{h} = \nabla \log h$, $\nabla \cdot \nabla \log h = \Delta \log h$, therefore we derive that

$$\Phi_X(h) = \frac{1}{2} g(t)^2 \|\nabla \log h\|^2$$

For the score matching operator associated with $\mathcal{L}_Y$, it can be expressed as,

$$
\begin{aligned}
\Phi_Y(h) &= \frac{\mathcal{L}_Y h}{h} - \mathcal{L}_Y \log h \\
&= \sum_{y \in \mathbb{X}} Q_s(y, y_s) \Big( \frac{h(x_t, y, t, s)}{h(x_t, y_s, t, s)} - \log h(x_t, y, t, s) \Big) \\
&= \sum_{y \in \mathbb{X}} Q_s(y, y_s) \Big( \frac{h(x_t, y, t, s)}{h(x_t, y_s, t, s)} - \log \frac{h(x_t, y, t, s)}{h(x_t, y_s, t, s)} \Big)
\end{aligned}
$$

where the last line follows since $\sum_{y \in \mathbb{X}} Q_s(y, y_s) = 0$.

In this case, the $\mathcal{L}_X$ related term $\mathcal{J}_X$ in $\mathcal{I}_{\mathrm{GDSM}}$ can be simplified as,

$$
\begin{aligned}
\mathcal{J}_X &= \frac{\mathcal{L}_X(p_{\boldsymbol{t}|0}/\beta_\theta)(\boldsymbol{x_t}, \boldsymbol{t})}{(p_{\boldsymbol{t}|0}/\beta_\theta)(\boldsymbol{x_t}, \boldsymbol{t})} - \mathcal{L}_X \log(p_{\boldsymbol{t}|0}/\beta_\theta)(\boldsymbol{x_t}, \boldsymbol{t}) \\
&= \mathop{\mathbb{E}}_{\boldsymbol{t}, p_0, p_{\boldsymbol{t}|0}} \Big[ \Phi_X \big( \frac{p_{\boldsymbol{t}|0}}{\beta_\theta} \big) \Big] \\
&= \mathop{\mathbb{E}}_{\boldsymbol{t}, p_0, p_{\boldsymbol{t}|0}} \Big[ \frac{1}{2} g(t)^2 \Big\| \nabla \log p(x_t, y_s, t, s \mid x_0, y_0) - \nabla \log \beta_\theta(x_t, y_s, t, s) \Big\|^2 \Big]
\end{aligned}
$$

Moreover, since $(X_t, Y_s)$ are conditionally independent given $(X_0, Y_0)$, we have that

$$
\begin{aligned}
\nabla \log p(x_t, y_s, t, s \mid x_0, y_0) &= \nabla \log \big( p(x_t, t \mid x_0, y_0) \cdot p(y_s, s \mid x_0, Y_0) \big) \\
&= \nabla \log p(x_t, t \mid x_0, y_0) + \nabla \log p(y_s, s \mid x_0, y_0) \\
&= \nabla \log p(x_t, t \mid x_0)
\end{aligned}
$$

which suggests that under our framework, the multimodal conditional score is identical to the unimodal conditional score. Using this, and set $s_\theta^X(x_t, y_s, t, s) = \nabla \log \beta_\theta(x_t, y_s, t, s)$, we have

$$\mathcal{J}_X = \mathop{\mathbb{E}}_{\substack{t, s, x_0, y_0 \sim p_0 \\ x_t, y_s \sim p_{t, s|0}}} \Big[ \frac{1}{2} g^2(t) \big\| s_\theta^X - \nabla \log p_t(x_t|x_0) \big\|^2 \Big]$$

Similarly, the $\mathcal{L}_Y$ related term $\mathcal{J}_Y$ in $\mathcal{I}_{\mathrm{GDSM}}$ can be simplified as,

$$
\begin{aligned}
\mathcal{J}_Y &= \mathop{\mathbb{E}}_{t,p_0,p_{t|0}} \left[ \Phi_Y\left(\frac{p_{t|0}}{\beta_\theta}\right) \right] \\
&= \mathop{\mathbb{E}}_{t,p_0,p_{t|0}} \left[ \sum_{y \in \mathbb{X}} Q_s(y,y_s) \left( \frac{p(x_t,y,t,s \mid x_0,y_0)}{p(x_t,y_s,t,s \mid x_0,y_0)} \cdot \frac{\beta_\theta(x_t,y_s,t,s)}{\beta_\theta(x_t,y,t,s)} - \log \frac{p(x_t,y,t,s \mid x_0,y_0)}{p(x_t,y_s,t,s \mid x_0,y_0)} + \log \frac{\beta_\theta(x_t,y,t,s)}{\beta_\theta(x_t,y_s,t,s)} \right) \right] \\
&= \mathop{\mathbb{E}}_{t,p_0} \left[ \int_{\mathbb{R}^d} \left( \sum_{y,y_s \in \mathbb{X}} Q_s(y,y_s) \left( p(x_t,y,t,s \mid x_0,y_0) \cdot \frac{\beta_\theta(x_t,y_s,t,s)}{\beta_\theta(x_t,y,t,s)} + p(x_t,y_s,t,s \mid x_0,y_0) \cdot \log \frac{\beta_\theta(x_t,y,t,s)}{\beta_\theta(x_t,y_s,t,s)} \right) \right) \mathrm{d}x_t \right] \\
&\qquad + \mathrm{const} \\
&= \mathop{\mathbb{E}}_{t,p_0} \left[ \sum_{y_s \in \mathbb{X}} \int_{\mathbb{R}^d} \left( \mathop{\mathbb{E}}_{y \sim p(x_t,\cdot,t,s|x_0,y_0)} Q_s(y,y_s) \left( \frac{\beta_\theta(x_t,y_s,t,s)}{\beta_\theta(x_t,y,t,s)} + \frac{p(x_t,y_s,t,s \mid x_0,y_0)}{p(x_t,y,t,s \mid x_0,y_0)} \cdot \log \frac{\beta_\theta(x_t,y,t,s)}{\beta_\theta(x_t,y_s,t,s)} \right) \right) \mathrm{d}x_t \right] \\
&\qquad + \mathrm{const}
\end{aligned}
$$

Now, we exchange variable $y_s$ and $y$, and thus the expression is rewritten to,

$$
\begin{aligned}
\mathcal{J}_Y &= \mathop{\mathbb{E}}_{t,p_0} \left[ \sum_{y \in \mathbb{X}} \int_{\mathbb{R}^d} \left( \mathop{\mathbb{E}}_{y_s \sim p(x_t,\cdot,t,s|x_0,y_0)} Q_s(y_s,y) \left( \frac{\beta_\theta(x_t,y,t,s)}{\beta_\theta(x_t,y_s,t,s)} - \frac{p(x_t,y,t,s \mid x_0,y_0)}{p(x_t,y_s,t,s \mid x_0,y_0)} \cdot \log \frac{\beta_\theta(x_t,y,t,s)}{\beta_\theta(x_t,y_s,t,s)} \right) \right) \mathrm{d}x_t \right] \\
&\qquad + \mathrm{const} \\
&= \mathop{\mathbb{E}}_{t,p_0,p_{t|0}} \left[ \sum_{y \in \mathbb{X}} Q_s(y_s,y) \left( \frac{\beta_\theta(x_t,y,t,s)}{\beta_\theta(x_t,y_s,t,s)} - \frac{p(x_t,y,t,s \mid x_0,y_0)}{p(x_t,y_s,t,s \mid x_0,y_0)} \cdot \log \frac{\beta_\theta(x_t,y,t,s)}{\beta_\theta(x_t,y_s,t,s)} \right) \right] + \mathrm{const}
\end{aligned}
$$

Using again the conditional independence of $(X_t, Y_s)$ given $(X_0, Y_0)$, we have that

$$
\begin{aligned}
\frac{p(x_t,y,t,s \mid x_0,y_0)}{p(x_t,y_s,t,s \mid x_0,y_0)} &= \frac{p(x_t,t \mid x_0,y_0)p(y,s \mid x_0,y_0)}{p(x_t,t \mid x_0,y_0)p(y_s,s \mid x_0,y_0)} \\
&= \frac{p(y,s \mid x_0,y_0)}{p(y_s,s \mid x_0,y_0)} = \frac{p(y,s \mid y_0)}{p(y_s,s \mid y_0)}
\end{aligned}
$$

which indicates again that the multimodal conditional score is identical to the unimodal conditional score. Set $s_\theta^Y(x_t,y_s,t,s)_y = \beta_\theta(x_t,y,t,s)/\beta_\theta(x_t,y_s,t,s)$, we finally arrive at that

$$
\mathcal{J}_Y = \mathop{\mathbb{E}}_{\substack{t,s,x_0,y_0 \sim p_0 \\ x_t,y_s \sim p_{t,s|0}}} \left[ \sum_{y \in \mathbb{X}} Q_s(y_s,y) \left( s_\theta^Y(x_t,y_s,t,s)_y - \frac{p(y,s \mid y_0)}{p(y_s,s \mid y_0)} \cdot \log s_\theta^Y(x_t,y_s,t,s)_y \right) \right] + \mathrm{const}
$$

Note that while the sum in $\mathcal{J}_Y$ is over $y \in \mathbb{X}$, in fact when $y = y_s$, the corresponding term is constant and has no contribution to the gradient of $\theta$, therefore we can instead only summing over $y \in \mathbb{X}$ and $y \neq y_s$, recovering the presented expressions in Prop. 1. This concludes the proof. □

### A.4. Proof of Proposition 2

We start by considering the function $u(x,y,t,s) = \mathbb{E}[h(X_t, Y_s) \mid X_0 = x, Y_0 = y]$ as test functions. We start by computing the generator of the forward process. For notational convenience, we use $\mathcal{L}^X$ and $\mathcal{L}^Y$ to denote the generators of $X_t$ and $Y_s$, respectively.

**Lemma 1** (Generator of the forward process). *Given $(X_t, Y_s)$ following dynamic (6), and a test function $u : \mathbb{R}^d \times \mathbb{X}$, we have that:*

$$
\mathcal{L}^X u = \langle \nabla_x u(x,y), f(x,t) \rangle + \frac{1}{2} g^2(t) \cdot \langle \nabla_x^2 u(x,y), I \rangle = f \cdot \nabla u + \frac{1}{2} g^2(t) \Delta u \tag{10}
$$

$$
\mathcal{L}^Y u = \sum_{\hat{y} \in \mathbb{X}} u(x,\hat{y}) Q_s(\hat{y},y) = Q_s^\top u(x,\cdot) \tag{11}
$$

*Proof.* We start by computing the generator of $Y_s$:

$$
\begin{aligned}
\mathcal{L}^Y u &= \lim_{\Delta s \to 0} \frac{1}{\Delta s} \cdot \mathbb{E}\Big[ u(X_t, Y_{s+\Delta s}) - u(X_t, Y_s) \mid X_t = x, Y_s = y \Big] \\
&= \lim_{\Delta s \to 0} \frac{1}{\Delta s} \cdot \sum_{\hat{y} \in \mathbb{X}} u(x, \hat{y}) \big( \mathbb{P}(Y_{s+\Delta s} = \hat{y} \mid Y_s = y) - \mathbb{P}(Y_s = \hat{y} \mid Y_s = y) \big) \\
&= \sum_{\hat{y} \in \mathbb{X}} u(x, \hat{y}) \lim_{\Delta s \to 0} \frac{1}{\Delta s} \big( \mathbb{P}(Y_{s+\Delta s} = \hat{y} \mid Y_s = y) - \mathbb{P}(Y_s = \hat{y} \mid Y_s = y) \big) \\
&= \sum_{\hat{y} \in \mathbb{X}} u(x, \hat{y}) Q_s(\hat{y}, y) = Q_s^\top u(x, \cdot)
\end{aligned}
$$

Similarly, for the generator of $X_t$,

$$
\begin{aligned}
\mathcal{L}^X u &= \lim_{\Delta t \to 0} \frac{1}{\Delta t} \mathbb{E}\Big[ u(X_{t+\Delta t}, Y_s) - u(X_t, Y_s) \mid X_t = x, Y_s = y \Big] \\
&= \lim_{\Delta t \to 0} \frac{1}{\Delta t} \cdot \mathbb{E}\Big[ \int_t^{t+\Delta t} \langle f(X_\tau, Y_s), \nabla_x u(X_\tau,, Y_s) \rangle + \frac{1}{2} \langle g^2(\tau) I, \nabla_x^2 u(X_\tau, Y_s) \rangle \mathrm{d}\tau \mid X_t = x, Y_s = y \Big] \\
&= \mathbb{E}\Big[ \langle f(X_t, Y_s), \nabla_x u(X_t,, Y_s) \rangle + \frac{1}{2} \langle g^2(t), \nabla^2 u(X_t, Y_s) \rangle \mid X_t = x, Y_s = y \Big] \\
&= (f \cdot \nabla u)(x, y) + \frac{1}{2} g^2(t) \Delta u(x, y)
\end{aligned}
$$

where we uses Ito's lemma and the fact that the martingale vanishes under the expectation. $\qquad \square$

We now compute the generator for the backward process:

$$
\begin{cases}
\mathrm{d}\overleftarrow{X}_t = -f(\overleftarrow{X}_t, T-t) + g^2(T-t)\nabla_x \log \overleftarrow{p}_{t,s}(\overleftarrow{X}_t, \overleftarrow{Y}_s)\mathrm{d}t + g(t)\mathrm{d}\overleftarrow{W}_t \\
\overleftarrow{Y}_s \sim \mathrm{CTMC}\left(\overleftarrow{Q}(\overleftarrow{X}_t, t, s)\right) \\
(\overleftarrow{X}_0, \overleftarrow{Y}_0) \sim p(x, y, T, T)
\end{cases}
\tag{12}
$$

where $\overleftarrow{p}_{t,s} = p_{T-t,T-s}$, $\overleftarrow{Q}(X_t, t, s)$ is a rate matrix with $y', y$ entry being $\frac{\overleftarrow{p}_{t,s}(X_t, y')}{\overleftarrow{p}_{t,s}(X_t, y)}(Q_{T-s})_{yy'}$. We denote $\mathcal{L}_{\leftarrow}^X$ and $\mathcal{L}_{\leftarrow}^Y$ as the generator of backward process $\overleftarrow{X}_t, \overleftarrow{Y}_s$ in (12) respectively.

**Lemma 2** (Generator of the backward process). *Given $X_t, Y_s$ following (12) and a test function $u : \mathbb{R}^d \times \mathbb{X}$, we have that:*

$$
\begin{aligned}
\mathcal{L}_{\leftarrow}^X u &= -\langle \nabla_x u(x, y), f(x, T-t) \rangle + \frac{1}{2} g^2(T-t)\langle \nabla_x^2 u(x, y), I \rangle - \langle \nabla_x u(x, y), g^2(T-t)\nabla_x \log \overleftarrow{p}_{t,s}(x, y) \rangle \\
\mathcal{L}_{\leftarrow}^Y u &= \overleftarrow{Q}(x, t, s)^\top u = \sum_{\hat{y} \in \mathbb{X}} u(x, \hat{y}) \cdot \overleftarrow{Q}(x, t, s)(\hat{y}, y)
\end{aligned}
$$

The proof is highly similar to that of Lem. 2, and thus we omit it here to avoid being repetitive. With $\mathcal{L}^X, \mathcal{L}^Y, \mathcal{L}_{\leftarrow}^X$ and $\mathcal{L}_{\leftarrow}^Y$ being computed, we can now derive the Fokker Planck equation for both the forward process $(X_t, Y_s)$ and backward process $(\overleftarrow{X}_t, \overleftarrow{Y}_s)$, as is given in Lem. 3 and Lem. 4.

**Lemma 3** (Fokker Planck Equation of Forward Process). *For $X_t, Y_s$ following (6), let $p(x, y, t, s)$ denotes the density of $X_t, Y_s$ at $X_t = x, Y_s = y$, we have that:*

$$
\partial_t p(x, y, t, s) = \mathcal{L}^{X,*} p(x, y, t, s), \quad \partial_s p(x, y, t, s) = \mathcal{L}^{Y,*} p(x, y, t, s)
\tag{13}
$$

*Where $\mathcal{L}^{X,*}, \mathcal{L}^{Y,*}$ represent the adjoint of $\mathcal{L}^X, \mathcal{L}^Y$ and:*

$$
\begin{aligned}
\mathcal{L}^{X,*} p(x, y, t, s) &= -\nabla \cdot (f(x, t) p(x, y, t, s)) + \frac{1}{2} g^2(t) \Delta p(x, y, t, s) \\
\mathcal{L}^{Y,*} p(x, \cdot, t, s) &= Q_s p(x, \cdot, t, s) = \sum_{\hat{y} \in \mathbb{X}} Q_s(y, \hat{y}) \cdot p(x, \hat{y}, t, s)
\end{aligned}
$$

**Lemma 4** (Fokker Planck Equation of Backward Process). *For $\overleftarrow{X}_t, \overleftarrow{Y}_s$ following (6), let $\overline{p}(x, y, t, s)$ denotes the density of $\overleftarrow{X}_t, \overleftarrow{Y}_s$) at $\overleftarrow{X}_t = x, \overleftarrow{Y}_s = y$, we have that:*

$$\partial_t \overline{p}(x, y, t, s) = \mathcal{L}^{X,*}_{\leftarrow} \overline{p}(x, y, t, s), \qquad \partial_s \overline{p}(x, y, t, s) = \mathcal{L}^{Y,*}_{\leftarrow} \overline{p}(x, y, t, s) \tag{14}$$

*where $\mathcal{L}^{X,*}_{\leftarrow}, \mathcal{L}^{Y,*}_{\leftarrow}$ represent the adjoint of $\mathcal{L}^X_{\leftarrow}, \mathcal{L}^Y_{\leftarrow}$ and:*

$$\mathcal{L}^{X,*}_{\leftarrow} \overline{p}(x, y, t, s) = \nabla \cdot (f(x, T - t)\overline{p}(x, y, t, s)) + \frac{1}{2}g^2(T - t)\Delta \overline{p}(x, y, t, s) - g^2(T - t)\nabla \cdot (\overline{p}(x, y, t, s)\nabla_x \log \overleftarrow{p}_{t,s}(x, y))$$

$$\mathcal{L}^{Y,*}_{\leftarrow} \overline{p}(x, \cdot, t, s) = \overleftarrow{Q}(x, t, s)\overline{p}(x, \cdot, t, s) = \sum_{\hat{y} \in \mathbb{X}} \overleftarrow{Q}(x, t, s)(y, \hat{y})\overline{p}(x, \hat{y}, t, s)$$

The computation central to the proof of Lem. 3 and Lem. 4 is the calculation of the adjoint operator, which can be done with standard techniques, such as integration by parts. The derivation of the Fokker Planck equations directly follows from definitions. Therefore, we also omit the proof here. With these results, we are now ready to show that $(X_t, Y_s)$ and $\overleftarrow{X}_t, \overleftarrow{Y}_s$ are time reversals of each other.

**Lemma 5** (Time Reversal). $\overline{p}(x, y, t, s) = \overleftarrow{p}_{t,s}(x, y) = p(x, y, T - t, T - s)$, where $p$ is the solution to the Fokker Planck equation of the forward process in (13), $\overline{p}$ is the solution to the Fokker Planck equation of the backward process in (14).

*Proof.* We will prove the result by showing that $\overline{p}(x, y, t, s) = p(x, y, T - t, T - s)$ satisfies the Fokker Planck equations given in (14). We start by showing the $\overleftarrow{X}_t$ related equation. Substituting in $p(x, y, T - t, T - s)$, we have that

$$\partial_t \overline{p}(x, y, t, s) = \partial_t(p(x, y, T - t, T - s)) = -\partial_t p(x, y, T - t, T - s) = -\mathcal{L}^{X,*} p(x, y, T - t, T - s)$$

where the last equality holds due to (13). On the right side of the equation, the expression can be simplified to:

$$\begin{aligned}
\mathcal{L}^{X,*}_{\leftarrow} \overline{p}(x, y, t, s) &= \mathcal{L}^{X,*}_{\leftarrow} p(x, y, T - t, T - s) \\
&= \nabla \cdot (f(x, T - t)p(x, y, T - t, T - s) + \frac{1}{2}g^2(T - t)\Delta p(x, y, T - t, T - s) \\
&\quad - g^2(T - t)\nabla \cdot (p(x, y, T - t, T - s)\nabla \log p(x, y, T - t, T - s)) \\
&= \nabla \cdot (f(x, T - t)p(x, y, T - t, T - s) + \frac{1}{2}g^2(T - t)\Delta p(x, y, T - t, T - s) \\
&\quad - g^2(T - t)\nabla \cdot \nabla p(x, y, T - t, T - s) \\
&= \nabla \cdot (f(x, T - t)p(x, y, T - t, T - s) - \frac{1}{2}g^2(T - t)\Delta p(x, y, T - t, T - s) \\
&= -\mathcal{L}^{X,*} p(x, y, T - t, T - s)
\end{aligned}$$

Therefore, we show that $\partial_t \overline{p}(x, y, t, s) = \mathcal{L}^{X,*}_{\leftarrow} \overline{p}(x, y, t, s)$ when $\overline{p}(x, y, t, s) = p(x, y, T - t, T - s)$. For the $\overleftarrow{Y}_s$ related equation, we have that

$$\partial_s \overline{p}(x, y, t, s) = \partial_s(p(x, y, T - t, T - s)) = -\partial_s p(x, y, T - t, T - s) = -\mathcal{L}^{Y,*} p(x, y, T - t, T - s)$$

Similarly, on the right size of the equation, we have,

$$\begin{aligned}
\mathcal{L}^{Y,*}_{\leftarrow} \overline{p}(x, y, t, s) &= \mathcal{L}^{Y,*}_{\leftarrow} p(x, y, T - t, T - s) \\
&= \sum_{\hat{y} \in \mathbb{X}} \overleftarrow{Q}(x, t, s)(y, \hat{y})p(x, \hat{y}, T - t, T - s) \\
&= \sum_{\hat{y} \in \mathbb{X}, \hat{y} \neq y} \overleftarrow{Q}(x, t, s)(y, \hat{y})p(x, \hat{y}, T - t, T - s) + \overleftarrow{Q}(x, t, s)(y, y)p(x, y, T - t, T - s) \\
&= \sum_{\hat{y} \in \mathbb{X}, \hat{y} \neq y} \overleftarrow{Q}(x, t, s)(y, \hat{y})p(x, \hat{y}, T - t, T - s) - \sum_{\hat{y} \in \mathbb{X}, \hat{y} \neq y} \overleftarrow{Q}(x, t, s)(\hat{y}, y)p(x, y, T - t, T - s)
\end{aligned}$$

where in the last step, we use that $\overleftarrow{Q}(x,t,s)(y,y) = -\sum_{\hat{y}\in\mathbb{X}, \hat{y}\neq y} \overleftarrow{Q}(x,t,s)(\hat{y},y)$. We perform such a simplification since we can only relate $\overleftarrow{Q}(x,t,s)$ and $Q_{T-s}$ on non-diagonal entries. Then, it holds that

$$
\begin{aligned}
\mathcal{L}_{\leftarrow}^{Y,*}\overline{p}(x,y,t,s) &= \sum_{\hat{y}\in\mathbb{X}, \hat{y}\neq y} Q_{T-s}(\hat{y},y) \cdot \frac{p(x,y,T-t,T-s)}{p(x,\hat{y},T-t,T-s)} p(x,\hat{y},T-t,T-s) \\
&\qquad - Q_{T-s}(y,\hat{y}) \cdot \frac{p(x,\hat{y},T-t,T-s)}{p(x,y,T-t,T-s)} p(x,y,T-t,T-s) \\
&\quad \sum_{\hat{y}\in\mathbb{X}, \hat{y}\neq y} Q_{T-s}(\hat{y},y) p(x,y,T-t,T-s) - Q_{T-s}(y,\hat{y}) p(x,\hat{y},T-t,T-s) \\
&= \sum_{\hat{y}\in\mathbb{X}} Q_{T-s}(\hat{y},y) p(x,y,T-t,T-s) - Q_{T-s}(y,\hat{y}) p(x,\hat{y},T-t,T-s) \\
&= p(x,y,T-t,T-s) \cdot \Big(\sum_{\hat{y}\in\mathbb{X}} Q_{T-s}(\hat{y},y)\Big) - \sum_{\hat{y}\in\mathbb{X}} Q_{T-s}(y,\hat{y}) p(x,\hat{y},T-t,T-s) \\
&= -\sum_{\hat{y}\in\mathbb{X}} Q_{T-s}(y,\hat{y}) p(x,\hat{y},T-t,T-s) \\
&= -\mathcal{L}^{Y,*} p(x,y,T-t,T-s)
\end{aligned}
$$

where in the derivation, we use the definition of $\overleftarrow{Q}(x,t,s)$ and the fact that $\sum_{\hat{y}\in\mathbb{X}} Q_{T-s}(\hat{y},y) = 0$. Therefore, we have also shown that $\partial_t \overline{p}(x,y,t,s) = \mathcal{L}_{\leftarrow}^{Y,*}\overline{p}(x,y,t,s)$ when $\overline{p}(x,y,t,s) = p(x,y,T-t,T-s)$. Together with the fact that the initial conditions are matched by construction, i.e.,

$$
\overline{p}(x,y,0,0) = p(x,y,T,T)
$$

we conclude the proof of the time-reversal argument, as well as [Prop. 2](#).

$\square$

# B. Design choice for Continuous-Discrete Multimodal Diffusion Models

In this section, we go deeper into the design choices of the newly proposed continuous-discrete multimodal diffusion models, including the choice for forward process, score parameterization, and sampling algorithms for such diffusion models.

## B.1. Choice of Forward Process

We consider the following specific choice of forward process for the Continuous-Discrete Diffusion Model, where we choose $X_t$ to be subjected to a time-rescaled Ornstein–Uhlenbeck process ([Song et al., 2020](#)) and $Y_s$ to be subjected to a masked discrete diffusion model ([Ou et al., 2025](#); [Sahoo et al., 2024](#); [Shi et al., 2024](#)). Both choices are effective when modeling unimodal distributions, prompting us to combine them for the design of multimodal diffusion models on their product space.

$$
\begin{cases}
\mathrm{d}X_t = -\beta_t X_t \mathrm{d}t + \sqrt{2\beta_t}\mathrm{d}B_t \\
Y_s \sim \mathrm{CTMC}(\sigma_s Q^{\mathrm{mask}}) \\
(X_0, Y_0) \sim p_{\mathrm{data}}(x,y)
\end{cases}
\tag{15}
$$

To define masked discrete diffusion models, we need to introduce an additional mask token $\mathbf{M}$ into the state space $\mathbb{X}$. Therefore, the transition rate matrix $Q^{\mathrm{mask}}$ is a transition matrix defined on the extended state space $\tilde{\mathbb{X}} = \mathbb{X} \cup \{\mathbf{M}\}$, given by

$$
Q^{\mathrm{mask}} = \begin{pmatrix}
-1 & 0 & \dots & 0 \\
0 & -1 & \dots & 0 \\
\vdots & \vdots & \ddots & \vdots \\
1 & 1 \dots & 1 & 0
\end{pmatrix}
$$

where the last row corresponds to $\mathbf{M}$.

Consider now the practical case, where $X_t \in \mathbb{R}^d$, and $Y_s$ consists of a sequence of tokens, i.e., $Y_s \in \mathbb{X}^n$. We represent $Y_s$ as $Y_s = y_1 \ldots y_i \ldots y_n$.

For pure masked discrete diffusion, it has been shown that its score has a special factorization (Ou et al., 2025). This enables users to parameterize only the probability of the clean data distribution conditioned on unmasked tokens, thereby representing the score. In the following, we demonstrate that this special structure is inherited in the continuous-discrete diffusion model when the discrete part is a masked discrete diffusion, which enables a more effective score parameterization, as well as a variance-reduced version of the training objective.

Now, consider $\boldsymbol{y} = y_1 \ldots y_i \ldots y_n$, and $\hat{\boldsymbol{y}} = y_1 \ldots \hat{y}_i \ldots y_n$, where $\boldsymbol{y}$ differs from $\hat{\boldsymbol{y}}$ only at the $i$-th position, $\hat{y}_i = \mathbf{M}$ while $y_i \neq \mathbf{M}$

**Proposition 3.** *Let $p(x, \boldsymbol{y}, t, s)$ be the density of (15) at $X_t = x, Y_s = \boldsymbol{y}_s$, then the discrete score has the following form*

$$\frac{p(X_t = x_t, Y_s = \boldsymbol{y}, t, s)}{p(X_t = x_t, Y_s = \hat{\boldsymbol{y}}, t, s)} = \frac{e^{-\overline{\sigma}_s}}{1 - e^{-\overline{\sigma}_s}} \mathbb{P}(Y_0^i = y_i \mid \boldsymbol{y}^{\mathrm{UM}}, X_t, t) \tag{16}$$

*where $\overline{\sigma}_s = \int_0^s \sigma(\tau) \mathrm{d}\tau$, $\boldsymbol{y}^{\mathrm{UM}}$ contains the unmasked tokens of $\boldsymbol{y}$.*

*Proof.* We begin by noting that, given the forward process defined in (15). We can solve the forward process analytically as follows:

$$\mathbb{P}(Y_s^i = y \mid A) = \begin{cases} e^{-\overline{\sigma}_s} \cdot \mathbb{P}(Y_0^i = y \mid A), & y \neq \mathbf{M} \\ 1 - e^{-\overline{\sigma}_s}, & y = \mathbf{M} \end{cases} \tag{17}$$

For any event $A$. We will use this fact several times in our proof. This is true since when $y \neq \mathbf{M}$,

$$\mathbb{P}(Y_s^i = y \mid A) = \mathbb{P}(Y_s^i = y \mid Y_0^i = y, A) \cdot \mathbb{P}(Y_0^i = y \mid A)$$

Our proof consists of two steps. First, we demonstrate that the diffusion process can be factored into a conditional probability and a time-dependent term. Secondly, we demonstrate that we can further simplify this conditional probability to contain only probabilities in terms of the clean data distribution.

**Step 1:** The discrete score in (16) is given by:

$$\frac{\mathbb{P}(X_t = x_t, Y_s^1 = y_1, \ldots, Y_s^i = y_i, \ldots, Y_s^n = y_n)}{\mathbb{P}(X_t = x_t, Y_s^1 = y_1, \ldots, Y_s^i = \mathbf{M}, \ldots, Y_s^n = y_n)}$$

To simplify the notation, we define the following notation $A_i = \{Y_s^k = y_k : k \neq i\} \cap \{X_t = x_t\}$. Then, using Bayes' rule, we can rewrite the discrete score as:

$$\frac{\mathbb{P}(Y_s^i = y_i \mid A_i)\mathbb{P}(A_i)}{\mathbb{P}(Y_s^i = \mathbf{M} \mid A_i)\mathbb{P}(A_i)} = \frac{\mathbb{P}(Y_s^i = y_i \mid A_i)}{\mathbb{P}(Y_s^i = \mathbf{M} \mid A_i)} = \frac{e^{-\overline{\sigma}_s}}{1 - e^{-\overline{\sigma}_s}} \cdot \mathbb{P}(Y_0^i = y_i \mid A_i)$$

**Step 2:** We now show that we can simplify $A_i$ by removing the conditioning on tokens that are masked. But firstly we do a simple calculation that will come in handy, given events $A, B, C$ one has that:

$$\mathbb{P}(A \mid B \cap C) = \frac{\mathbb{P}(A \cap B \cap C)}{\mathbb{P}(B \cap C)} = \frac{\mathbb{P}(A \cap B \mid C)\mathbb{P}(C)}{\mathbb{P}(B \cap C)} = \frac{\mathbb{P}(C \mid A \cap B)\mathbb{P}(A \cap B)\mathbb{P}(C)}{\mathbb{P}(C)\mathbb{P}(B \cap C)}$$
$$= \frac{\mathbb{P}(A \cap B)\mathbb{P}(C \mid A \cap B)}{\mathbb{P}(B))\mathbb{P}(C \mid B)} = \frac{\mathbb{P}(A \mid B)\mathbb{P}(C \mid A \cap B)}{\mathbb{P}(C \mid B)}$$

Now, assume that $l$ is a position such that $y_l = \mathbf{M}$. We denote $A_i^l = \{Y_s^k = y_k : k \neq i, l\} \cap \{X_t = x_t\}$ the event given by

the remaining tokens. We can then use the calculation above to write:

$$\mathbb{P}(Y_0^i = y_i \mid A_i) = \mathbb{P}(Y_0^i = y_i \mid A_i^l, Y_s^l = M)$$
$$= \frac{\mathbb{P}(\{Y_0^i = y_i\} \cap A_i^l \cap \{Y_s^l = M\})}{\mathbb{P}(A_i^l \cap \{Y_s^l = M\})})$$
$$= \mathbb{P}(Y_0^i = y_i \mid A_i^l) \cdot \frac{\mathbb{P}(Y_s^l = \mathbf{M} \mid \{Y_0^i = y_i\} \cap A_i^l)}{\mathbb{P}(Y_s^l = \mathbf{M} \mid A_i^l)}$$
$$= \mathbb{P}(Y_0^i = y_i \mid A_i^l)$$

where in the last step, we used (17). The computation above shows that we can remove any events relevant to masked tokens, and repeating this process yields the result that the event being conditioned on only contains the unmasked part. This finishes the proof. □

With Prop. 3, the score entropy in the training objective described in Prop. 1 can be further simplified to cross-entropy loss, as is widely discussed in the literature of masked discrete diffusion model training (Ou et al., 2025; Sahoo et al., 2024; Shi et al., 2024). Prop. 3 also implies that when designing the score network backbone, the discrete score does not need the input of time $s$. However, it's still dependent on the time $t$ of the continuous modality. This is a notable difference from the pure masked discrete diffusion models considered in the literature (Ou et al., 2025; Nie et al., 2025a).

## B.2. Noisy Guidance

In Alg. 1, we present the detailed algorithm for using Noisy guidance for continuous score. In Alg. 2, we present the detailed algorithm for using Noisy guidance for discrete score, whose computation is not mentioned in the main text. Note that in Alg. 2, instead of directly doing geometric average of $s_\theta^{\mathrm{uncond},y}$ and $s_\theta^{\mathrm{cond},y}$ as is suggested in Nisonoff et al. (2024), we compute a arthimetic average of the corresponding logits, then use softmax to calculate the actual guided score. Such a practice is considered in Chang et al. (2022), and recently it has been shown to have theoretical advantages in Ye et al. (2025); Rojas et al. (2025a).

---

**Algorithm 1** Noisy Guidance for continuous score

---

**Require:** $x_t, t$ : noisy image with noise level, model: $s_\theta(x_t, y_s, t, s, \omega)$, $y_0$ : clean text, $\omega$ : Guidance Strength, $\sigma$ : Conditioning Noise Level
**Ensure:** $s_\theta^x$ : Guided continuous score
1: $y_\sigma^{\mathrm{noisy}} \sim p_{\sigma|0}(y|y_0)$
2: $s_\theta^{\mathrm{uncond},x} \leftarrow s_\theta(x, y_\sigma^{\mathrm{noisy}}, t, \sigma, \omega_t)$, $s_\theta^{\mathrm{cond},x} \leftarrow s_\theta(x_t, y_0, t, 0, \omega_t)$
3: $s_\theta^x \leftarrow \omega \cdot s_\theta^{\mathrm{cond},x} + (1-\omega) \cdot s_\theta^{\mathrm{uncond},x}$
4: **return** $s_\theta^x$

---

**Algorithm 2** Noisy Guidance for discrete score

---

**Require:** $y_s, s$ : noisy text with noise level, model: $s_\theta(x_t, y_s, t, s, \omega)$, $x_0$ : clean image, $\omega$ : Guidance Strength, $\sigma$ : Conditioning Noise Level
**Ensure:** $s_\theta^y$ : Guided discrete score
1: $x_\sigma^{\mathrm{noisy}} \sim p_{\sigma|0}(x|x_0)$
2: $s_\theta^{\mathrm{uncond},y} \leftarrow s_\theta(x_\sigma^{\mathrm{noisy}}, y_s, \sigma, s, \omega_s)$, $s_\theta^{\mathrm{cond},y} \leftarrow s_\theta(x_0, y_s, 0, s, \omega_s)$
3: $s_\theta^{\mathrm{uncond},y} = \mathrm{softmax}(\ell^{\mathrm{uncond}})$, $s_\theta^{\mathrm{cond},y} = \mathrm{softmax}(\ell^{\mathrm{cond}})$
4: $s_\theta^y \leftarrow \mathrm{softmax}\left(\omega \cdot \ell^{\mathrm{cond}} + (1-\omega) \cdot \ell^{\mathrm{uncond}}\right)$
5: **return** $s_\theta^y$

---

## B.3. Samplers

For inference of continuous-discrete multimodal diffusion models, we consider the following samplers. For conditional generation of discrete or continuous modality, see Alg. 3 and Alg. 4. Among all the pseudo code, we set times =

$(N - i)/N$, $i = 0, 1, \ldots, N$ as the inference times and $\mathrm{d}t = -1/N$ as the time steps, where $N$ is the number of total inference/discretization steps. We use $\tau$-leaping for discrete modality sampling and Heun's method for continuous modality sampling. All the inference algorithms are written with the presence of guidance and guidance intervals.

---

**Algorithm 3** Discrete Sampler with $\tau$-leaping

**Require:** $N$ : Number of steps, $\omega$ : Guidance Strength
1: $[a, b]$ : Guidance Interval, model : $s_\theta(x_t, y_s, t, s, \omega)$
2: $x$ : a clean image condition
**Ensure:** $y_0 \sim p_{\text{data}}(\cdot | x)$
3: $y_t \leftarrow [\mathbf{M}, \ldots, \mathbf{M}]$
4: **for** $t$ in times **do**
5: $\quad \omega_t = \omega$ if $t \in [a, b]$ else 1.
6: $\quad s_\theta^x, s_\theta^y \leftarrow s_\theta(x, y_t, 0, t, \omega_t)$
7: $\quad y_t \leftarrow \tau\text{-leaping}(s_\theta^y, y_t, t, |\mathrm{d}t|)$
8: **end for**
9: **return** $y_0$

---

**Algorithm 4** Continuous Sampler with Heun's method

**Require:** $N$ : Number of steps, $\omega$ : Guidance Strength
1: $[a, b]$ : Guidance Interval, model : $s_\theta(x_t, y_s, t, s, \omega)$
2: $y$ : A clean text condition
**Ensure:** $x_0 \sim p_{\text{data}}(\cdot | y)$
3: $x_t \leftarrow \mathcal{N}(0, I)$
4: **for** $t$ in times **do**
5: $\quad \omega_t = \omega$ if $t \in [a, b]$ else 1.
6: $\quad s_\theta^x, s_\theta^y \leftarrow s_\theta(x_t, y, 0, t, \omega_t)$
7: $\quad v_{\text{old}} = f(x_t, t) - \frac{1}{2} g^2(t) s_\theta^x$
8: $\quad \hat{x} \leftarrow x_t + v_{\text{old}} \mathrm{d}t, \quad \hat{s}_\theta^x, \hat{s}_\theta^y \leftarrow s_\theta(\hat{x}, y, t + \mathrm{d}t, 0, \omega_t)$
9: $\quad v_{\text{new}} = f(\hat{x}_t, t) - \frac{1}{2} g^2(t) \hat{s}_\theta^x$
10: $\quad x_t \leftarrow x_t + \frac{1}{2} \cdot (v_{\text{old}} + v_{\text{new}}) \mathrm{d}t$
11: **end for**
12: **return** $x_0$

---

We describe one multimodal sampler for the joint generation with a continuous-discrete multimodal diffusion model in Alg. 5. Essentially, Alg. 5 combines $\tau$-leaping for discrete modality and Heun's method for continuous modality, each depicted in Alg. 3 and Alg. 4. However, these choices are selected without being heavily optimized to tailor to this case, and potentially, there exist much more effective and efficient samplers. For example, note that Heun's method is a second-order ODE sampler, while $\tau$-leaping is usually considered to be a first-order CTMC sampling algorithm (Ren et al., 2024). This means that the discrete score obtained at the mid-point $\hat{x}$ during the inference step of Heun's method is not being used, which causes a waste of computation. A potential way to improve is to replace the first-order discrete sampler with a second-order variant, such as the $\theta$-Trap algorithm introduced in Ren et al. (2025a). We leave this direction for future investigation.

---

**Algorithm 5** Multimodal Sampler with $\tau$-leaping and Heun's Method

**Require:** $N$ : Number of steps, $\omega$ : Guidance Strength, $[a, b]$ : Guidance Interval, model : $s_\theta(x_t, y_s, t, s, \omega)$
**Ensure:** $x_0, y_0 \sim p_{\text{data}}$
1: $x_t \leftarrow \mathcal{N}(0, I), y_t \leftarrow [\mathbf{M}, \ldots, \mathbf{M}]$
2: **for** $t$ in times **do**
3: $\quad \omega_t = \omega$ if $t \in [a, b]$ else 1.
4: $\quad s_\theta^x, s_\theta^y \leftarrow s_\theta(x_t, y_t, t, t, \omega_t), \quad v_{\text{old}} = f(x_t, t) - \frac{1}{2} g^2(t) s_\theta^x$
5: $\quad \hat{x} \leftarrow x_t + v_{\text{old}} \mathrm{d}t$
6: $\quad \hat{s}_\theta^x, \hat{s}_\theta^y \leftarrow s_\theta(\hat{x}, y_t, t + \mathrm{d}t, t, \omega_t), \quad v_{\text{new}} = f(\hat{x}_t, t) - \frac{1}{2} g^2(t) \hat{s}_\theta^x$
7: $\quad x_t \leftarrow x_t + \frac{1}{2} \cdot (v_{\text{old}} + v_{\text{new}}) \mathrm{d}t, \quad y_t \leftarrow \tau\text{-leaping}(s_\theta^y, y_t, t, |\mathrm{d}t|)$
8: **end for**
9: **return** $x_0, y_0$

---

# C. Experimental Details on Text-Image Generation

## C.1. Choice of Forward Process

We consider the same forward process discussed in App. B.1, with $\beta_t$ and $\sigma_s$ given as

$$\beta_t = 500 \cdot (\sqrt{\beta_{\text{start}}}(1 - t) + t\sqrt{\beta_{\text{end}}})^2$$

$$\sigma_s = \frac{1 - \delta}{1 - (1 - \delta)s}$$

$$\beta_{\text{start}} = 0.00085, \quad \beta_{\text{end}} = 0.0120, \quad \delta = 10^{-5}$$

*Table 4.* Hyperparameters for inference of different tasks

| Parameter | text to image | image to text | joint |
|---|---|---|---|
| Number of Steps | 50 | 50 | 50 |
| Guidance Scale | 5.0 | 1.0 | 5.0 |
| Guidance Interval | $[0.3, 0.8]$ | - | $[0.3, 0.8]$ |
| Condition Noise Level | 0.77 | - | 1.0 |
| Early Stopping | $10^{-5}$ | $10^{-5}$ | $10^{-5}$ |

*Table 5.* Model hyperparameters

| parameter | value |
|---|---|
| patch size | 2 |
| joint depth | 8 |
| text depth | 6 |
| image depth | 6 |
| dim text | 1024 |
| dim image | 1024 |
| dim joint attention | 1024 |
| QK RMSnorm | true |
| dimension per head | 64 |
| number of heads | 8 |

*Table 6.* Training Hyperparameters

| Parameter | Stage 1 | Stage 2 | Stage 3 |
|---|---|---|---|
| Num Itr | 600K | 200K | 140K |
| EMA-$\beta$ | .99999 | .9999 | .9999 |
| Batch Size | 256 | 512 | 512 |
| Optimizer | AdamW | AdamW | AdamW |
| Learning Rate | 2e-4 | 2e-4 | 2e-4 |
| Adam-$\beta$'s | [.9, .9] | [.9, .9] | [.9, .9] |
| Weight Decay | 0.03 | 0.03 | 0.03 |

## C.2. Training Strategy

We divide our training into several stages. This is a standard practice for training vision-language models. In text-to-image diffusion models, a pretrained text encoder is used to achieve alignment between the text semantics and the image features. Popular choices in the literature are using CLIP or T5 as text encoders (Esser et al., 2024). However, in our use case, we require training on masked text, for which the availability of pretrained encoders is limited. For this reason, we decided not to use a pretrained text encoder. This has the advantage that we don't rely on any pretraining, which reduces the computational requirements of our model.

**Stage 1: Text-image Alignment** During this stage, we train both the joint embedding and the continuous decoder. We allow noisy text to be received as input to our model, meaning that we train on all possible combinations of $s$ and $t$, but without worrying about the text prediction task. We present all training hyperparameters across all stages below.

**Stage 2: Text prediction and Image Improvement** In this second stage, we freeze the joint embedding. We found that by doing so, we can simplify the training. The joint embedding is now capable of generating meaningful latent representations, which can then be used to predict clean text from masked tokens and a latent image representation.

**Stage 3: Multimodal Generation** Finally, we train both the image and text decoders. This is useful because the image decoder hasn't been trained specifically to predict from the frozen joint embeddings. Training the text decoder is not necessary, but we can get some extra training time by doing so. After this stage, our model is now capable of performing all tasks.

**Optional - Stage 4: Fine-tuning on downstream Tasks** When necessary, our models can be fine-tuned on downstream tasks to improve the performance.

## C.3. Sampling

For sampling, we use the samplers described in Alg. 3, Alg. 4, and Alg. 5, where we do not use guidance for the discrete component. Our default values for sampling are presented in Table 4.

## C.4. Hyperparamters

We include the network and training hyperparameters in table 5 and 6 respectively. The total model contains 578M parameters and the joint embedding plus a single modality is about 481M.

## D. Experimental Details on Mixed-type Tabular Data Synthesis

### D.1. Choice of Forward Process

We consider the same forward process discussed in App. B.1, with $\beta_t$ and $\sigma_s$ given as

$$\beta_t = \beta_{\text{start}}(1 - t) + t\beta_{\text{end}}$$
$$\sigma_s = \frac{1 - \delta}{1 - (1 - \delta)s}$$
$$\beta_{\text{start}} = 0.1, \quad \beta_{\text{end}} = 20 \quad \delta = 10^{-5}$$

### D.2. Detailed description of datasets

We evaluate our model on six tabular datasets (https://archive.ics.uci.edu/datasets): Adult, Default, Shoppers, Magic, Beijing, and News. Beijing and News datasets are designed for regression task while the other four datasets are for the classification task.

Table 7. Statistics for the tabular datasets.

| Dataset | #Rows | #Numerical | #Categorical | #Training | #Test | Task |
|---|---|---|---|---|---|---|
| Adult | 48,842 | 6 | 9 | 32,561 | 16,281 | Classification |
| Default | 30,000 | 14 | 11 | 27,000 | 3,000 | Classification |
| Shoppers | 12,330 | 10 | 8 | 11,097 | 1,233 | Classification |
| Magic | 19,019 | 10 | 1 | 17,117 | 1,902 | Classification |
| Beijing | 41,757 | 7 | 5 | 37,581 | 4,176 | Regression |
| News | 39,644 | 46 | 2 | 35,679 | 3,965 | Regression |

### D.3. Model architecture and training details

The embedding for every numerical feature in the data is a summation of its type embedding and scale embedding. All numerical values share the same type embedding, which is a look-up table of size number of numerical features by the hidden dimension. Each numerical value is passed through a 3-layer MLP that expands a single numerical value to an embedding vector with the size of the hidden dimension. Categorical features in the data are individually embedded through a list of look-up embedding tables. The look-up embedding table has the size of the number of categories + 1 (with one extra mask token) by the hidden dimension. Then all the categorical look-up embedding tables are concatenated and treated as the categorical embedding for this dataset. The building block of our model is adopted from DiT (Peebles & Xie, 2023). The sinusoidal timestep is passed through a 2-layer MLP before input into the DiT blocks. After adding the integer positional embedding to the embedding, numerical embeddings and categorical embeddings are concatenated and input into DiT blocks. We used 4 DiT blocks with hidden dimension = 24 and number of heads = 4. The final layer splits the output into the numerical latent and a list of individual categorical latent. The latent vectors are passed into 3-layer MLPs to obtain the corresponding scores.

The noise perturbation is the variance preserving (VP) SDE. The training loss a weighted summation of the score matching loss for numerical features and score entropy loss for categorical features. The weighting parameter is chosen to balance the numerical and discrete loss. The optimizer is AdamW with learning rate = $10^{-3}$, weight decay = 0.03, $\beta = (0.9, 0.9)$. A linear rate warm-up scheduler is used with warmup steps = 200. The training batch size is 2048. We used EMA model for final evaluation. During sampling, we use Euler method for the continuous diffusion and tau-leaping for the discrete diffusion.

## D.4. Evaluation

We compare our model with five most recent generative models that are specifically designed to operate on tabular data: GOOGLE (Liu et al., 2023), StaSy (Kim et al., 2022), TabDDPM (Kotelnikov et al., 2023), CoDi (Lee et al., 2023), and TABSYN (Zhang et al., 2023). GOOGLE is a VAE-based method while the other four are all diffusion-based methods.

- **Shape** is a metric computed via the Kolmogorov-Smirnov Test between continuous distributions and Total Variation Distance between the probabilities for categorical values, measures, and compares the column-wise density between real and synthetic data.

- **Trend** is a metric that captures pair-wise column correlation by computing Pearson correlation for numerical columns, contingency similarity for categorical columns, and contingency similarity between bucketed numerical values and categorical values.

- **MLE** is the testing accuracy of the classification or regression task on real data after training an XGBoost Classifier or an XGBoost Regressor on the synthetic tabular data. For detailed training and optimization pipelines of MLE metric, please refer to the standardized pipeline proposed in Zhang et al. (2023).

- $\alpha$-**precision** evaluates if the synthetic data are from the same distribution as real-world data.

- $\beta$-**recall** quantifies whether the synthetic data can cover the entire distribution of the real data.

*Table 8.* Performance on the **Shape** metric in percentage (%). Higher values indicate better performance. Best performance in **bold**. Second best in underline.

| Methods | #Parameters | Adult | Default | Shoppers | Magic | Beijing | News |
|---|---|---|---|---|---|---|---|
| GOOGLE | $\sim$ 5.6M | 83.03 | 82.98 | 77.67 | 98.10 | 83.07 | 74.68 |
| STaSy | $\sim$ 10.3M | $88.71_{\pm 0.06}$ | $94.23_{\pm 0.06}$ | $90.63_{\pm 0.09}$ | $93.71_{\pm 0.13}$ | $93.29_{\pm 0.03}$ | $93.11_{\pm 0.03}$ |
| CoDi | $\sim$ 25.0M | $78.62_{\pm 0.06}$ | $84.23_{\pm 0.07}$ | $68.16_{\pm 0.05}$ | $88.44_{\pm 0.26}$ | $83.06_{\pm 0.02}$ | $67.73_{\pm 0.04}$ |
| TabDDPM | $\sim$ 11.7M | $98.25_{\pm 0.03}$ | $98.43_{\pm 0.08}$ | $97.28_{\pm 0.13}$ | $98.99_{\pm 0.09}$ | $\underline{98.70}_{\pm 0.03}$ | $21.25_{\pm 0.01}$ |
| TABSYN | $\sim$ 10.7M | $\underline{99.42}_{\pm 0.06}$ | $\underline{99.15}_{\pm 0.04}$ | $\mathbf{98.57}_{\pm 0.24}$ | $\mathbf{99.12}_{\pm 0.09}$ | $\mathbf{98.88}_{\pm 0.05}$ | $\underline{98.36}_{\pm 0.04}$ |
| TABSYN (reproduced) | $\sim$ 10.7M | $99.29_{\pm 0.06}$ | $97.12_{\pm 0.09}$ | $98.36_{\pm 0.10}$ | $\underline{99.02}_{\pm 0.10}$ | $96.35_{\pm 0.10}$ | $98.09_{\pm 0.03}$ |
| Our model | $\sim$ **64K** | $\mathbf{99.47}_{\pm 0.04}$ | $\mathbf{99.36}_{\pm 0.09}$ | $\underline{98.50}_{\pm 0.07}$ | $98.96_{\pm 0.16}$ | $97.94_{\pm 0.06}$ | $96.80_{\pm 0.05}$ |

*Table 9.* Performance on the $\alpha$-**precision** metric in percentage (%). Higher values indicate better performance. Best performance in **bold**. Second best in underline.

| Methods | #Parameters | Adult | Default | Shoppers | Magic | Beijing | News |
|---|---|---|---|---|---|---|---|
| GOOGLE | $\sim$ 5.6M | 50.68 | 68.89 | 86.95 | 90.88 | 88.81 | 86.41 |
| STaSy | $\sim$ 10.3M | $82.87_{\pm 0.26}$ | $90.48_{\pm 0.11}$ | $89.65_{\pm 0.25}$ | $86.56_{\pm 0.19}$ | $89.16_{\pm 0.12}$ | $94.76_{\pm 0.33}$ |
| CoDi | $\sim$ 25.0M | $77.58_{\pm 0.45}$ | $82.38_{\pm 0.15}$ | $94.95_{\pm 0.35}$ | $85.01_{\pm 0.36}$ | $98.13_{\pm 0.38}$ | $87.15_{\pm 0.12}$ |
| TabDDPM | $\sim$ 11.7M | $96.36_{\pm 0.20}$ | $97.59_{\pm 0.36}$ | $88.55_{\pm 0.68}$ | $98.59_{\pm 0.17}$ | $97.93_{\pm 0.30}$ | $0.00_{\pm 0.00}$ |
| TABSYN | $\sim$ 10.7M | $\mathbf{99.52}_{\pm 0.10}$ | $\underline{99.26}_{\pm 0.27}$ | $\underline{99.16}_{\pm 0.22}$ | $\mathbf{99.38}_{\pm 0.27}$ | $98.47_{\pm 0.10}$ | $\underline{96.80}_{\pm 0.25}$ |
| TABSYN (reproduced) | $\sim$ 10.7M | $99.32_{\pm 0.22}$ | $95.57_{\pm 0.33}$ | $\mathbf{99.22}_{\pm 0.31}$ | $\underline{99.21}_{\pm 0.27}$ | $\mathbf{98.87}_{\pm 0.15}$ | $96.30_{\pm 0.28}$ |
| Our model | $\sim$ **64K** | $\underline{99.47}_{\pm 0.17}$ | $\mathbf{99.47}_{\pm 0.21}$ | $98.78_{\pm 0.42}$ | $98.75_{\pm 0.36}$ | $\underline{98.49}_{\pm 0.24}$ | $\mathbf{97.47}_{\pm 0.27}$ |

*Table 10.* Performance on the $\beta$-**recall** metric in percentage (%). Higher values indicate better performance. Best performance in **bold**. Second best in underline.

| Methods | #Parameters | Adult | Default | Shoppers | Magic | Beijing | News |
|---|---|---|---|---|---|---|---|
| GOOGLE | $\sim$ 5.6M | 8.80 | 14.38 | 9.79 | 9.88 | 19.87 | 2.03 |
| STaSy | $\sim$ 10.3M | $29.21_{\pm 0.34}$ | $39.31_{\pm 0.39}$ | $37.24_{\pm 0.45}$ | $\mathbf{53.97}_{\pm 0.57}$ | $54.79_{\pm 0.18}$ | $39.42_{\pm 0.32}$ |
| CoDi | $\sim$ 25.0M | $9.20_{\pm 0.15}$ | $19.94_{\pm 0.22}$ | $20.82_{\pm 0.23}$ | $\underline{50.56}_{\pm 0.31}$ | $52.19_{\pm 0.12}$ | $34.40_{\pm 0.30}$ |
| TabDDPM | $\sim$ 11.7M | $47.05_{\pm 0.25}$ | $47.83_{\pm 0.35}$ | $47.79_{\pm 0.25}$ | $48.46_{\pm 0.42}$ | $\underline{56.92}_{\pm 0.13}$ | $0.00_{\pm 0.00}$ |
| TABSYN | $\sim$ 10.7M | $47.56_{\pm 0.22}$ | $\underline{48.00}_{\pm 0.35}$ | $\underline{48.95}_{\pm 0.28}$ | $48.03_{\pm 0.23}$ | $55.84_{\pm 0.19}$ | $\mathbf{45.04}_{\pm 0.34}$ |
| TABSYN (reproduced) | $\sim$ 10.7M | $\underline{47.75}_{\pm 0.21}$ | $42.95_{\pm 0.30}$ | $47.57_{\pm 0.44}$ | $47.92_{\pm 0.28}$ | $49.72_{\pm 0.27}$ | $44.37_{\pm 0.22}$ |
| Our model | $\sim$ **64K** | $\mathbf{49.65}_{\pm 0.26}$ | $\mathbf{48.29}_{\pm 0.32}$ | $\mathbf{51.25}_{\pm 0.50}$ | $47.66_{\pm 0.38}$ | $\mathbf{57.44}_{\pm 0.20}$ | $\underline{44.58}_{\pm 0.27}$ |

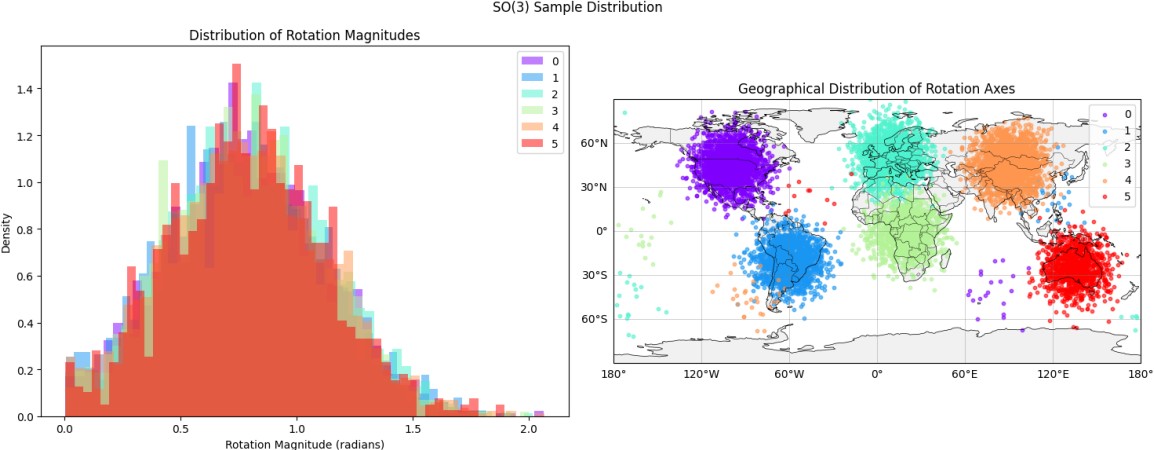

*Figure 5.* Visual representation of ground truth labeled Riemannian data.

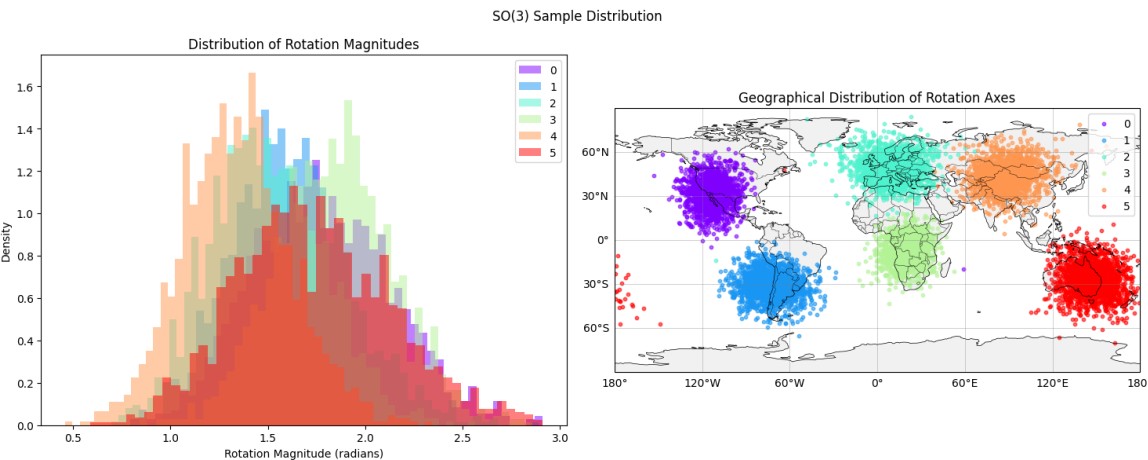

*Figure 6.* Peformance of inferring discrete label based on $\mathsf{SO}(3)$ data.

## E. Additional Experiment on Riemannian-Discrete Multimodal Diffusion Model

In this section, we demonstrate another application of our proposed multimodal diffusion model framework by focusing on the combination of Riemannian and discrete diffusion models on the state space $\mathcal{M} \times \mathbb{X}$, where $\mathcal{M}$ is a Riemannian manifold and $\mathbb{X}$ is a finite state space. We will introduce the method and validate it on a toy example consisting of synthetic data on $\mathsf{SO}(3) \times \mathbb{X}$.

### E.1. Riemannian-Discrete Multimodal Diffusion Model

We consider the setting where the target data distribution $p_{\text{data}}(x, y)$ is defined on $\mathsf{SO}(3) \times \mathbb{X}$, where $x \in \mathsf{SO}(3)$ and $y$ is a discrete label in $\mathbb{X}$. Since $\mathsf{SO}(3)$ is a compact manifold, we choose the following as the forward process,

$$\begin{cases} \mathrm{d}X_t = \mathrm{d}B_t^{\mathcal{M}} \\ Y_s \sim \text{CTMC}(Q_s) \\ (X_0, Y_0) \sim p_{\text{data}}(x, y) \end{cases} \tag{18}$$

where $Q_s = \sigma_s Q^{\text{mask}}$ is the same design choice as in (15), $\mathrm{d}B_t^{\mathcal{M}}$ is a Brownian Motion on $\mathsf{SO}(3)$. Note that the stationary distribution of (18) is $\text{Haar}(\mathsf{SO}(3)) \times \delta_{\mathbf{M}}$, where $\text{Haar}(\mathsf{SO}(3))$ is the Haar measure on $\mathsf{SO}(3)$, a generalized notion of uniform distribution. Following a similar derivation as is presented in the paper, we can derive its backward process,

$$\begin{cases} \mathrm{d}X_t = \nabla \log p(X_t, Y_s, T - t, T - s) + \mathrm{d}B_t^{\mathcal{M}} \\ Y_s \sim \mathrm{CTMC}(\overline{Q}(X_t, t, s) \end{cases} \tag{19}$$

where the gradient $\nabla$ is the Riemannian gradient, and $\overline{Q}(X_t, T - t, T - s)$ is defined for $y \neq \hat{y}$,

$$\overline{Q}(x, t, s)(\hat{y}, y) = \frac{p(x, \hat{y}, T - t, T - s)}{p(x, y, T - t, T - s)} Q_{T-s}(y, \hat{y})$$

**Axis-angle parametrization.** We represent elements of $\mathsf{SO}(3)$ using the axis-angle parametrization. We introduce it briefly here. One can show that any element of $S\mathsf{SO}(3)$ can be written as $\exp(\theta K)$ where:

$$K = a \begin{pmatrix} 0 & 0 & 0 \\ 0 & 0 & -1 \\ 0 & 1 & 0 \end{pmatrix} + b \begin{pmatrix} 0 & 0 & 1 \\ 0 & 0 & 0 \\ -1 & 0 & 0 \end{pmatrix} + c \begin{pmatrix} 0 & -1 & 0 \\ 1 & 0 & 0 \\ 0 & 0 & 0 \end{pmatrix}$$

and $(a, b, c) \in \mathbb{S}^2$ is a vector on the sphere, $\theta \in \mathbb{R}_+$. The representation $((a, b, c), \theta)$ is called the axis-angle representation.

**Dataset of the toy problem** We consider a simple toy example of labeled data on $\mathsf{SO}(3)$, consisting of Gaussian mixtures, where each mode corresponds to a unique label. To create the problem, we write elements $(a, b, c) \in \mathbb{S}^2$ in spherical coordinates; in this way, only two angles need to be parameterized. We then generate a Gaussian mixture on the space of these angles. Additionally, we use a von Mises random variable for $\theta$. We present the Python code used to generate the dataset in Listing E.1 and a visualization of the axis and angles in Figure 5.

As observed in Fig. 5, we have assigned labels to different geographical locations and assigned them to distinct modes on the map.

**Training strategy.** We train a simple MLP using a similar strategy as the text-image model. We first train a label to $\mathsf{SO}(3)$ model and add the discrete capabilities in a second phase. To achieve this, we utilize the generalized denoising score matching loss $\mathcal{I}_{\mathrm{GDSM}}$, as described in the main paper, which is derived from the generator computed based on the chosen forward process. We find that this training strategy is generally robust.

**Results.** We present samples generated by our method using guidance $w = 4$ in Figure 6, we see that our method can properly recover the data distribution. We also show the unconditional generation in Figure 7. We demonstrate that our method and training strategy can generalize to other data modalities.

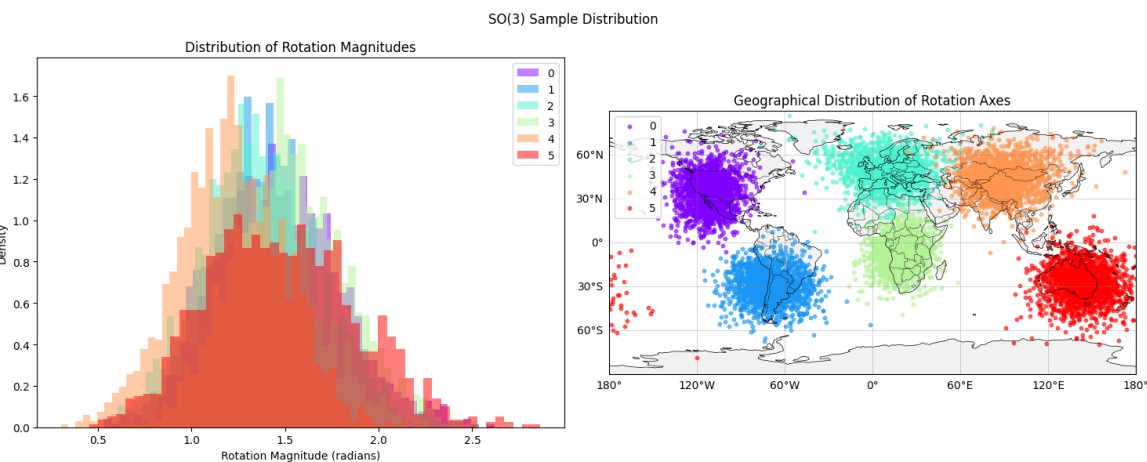

*Figure 7.* Performance of joint generation of the labeled Riemannian data.

```python
1   def sample_continental_so3(n_samples, magnitude_kappa=9):
2       continents = {
3           'North_America': {'lat': 45, 'lon': -100, 'weight': 0.2},
4           'South_America': {'lat': -20, 'lon': -60, 'weight': 0.15},
5           'Europe': {'lat': 50, 'lon': 10, 'weight': 0.2},
6           'Africa': {'lat': 0, 'lon': 20, 'weight': 0.15},
7           'Asia': {'lat': 45, 'lon': 90, 'weight': 0.2},
8           'Australia': {'lat': -25, 'lon': 135, 'weight': 0.1}
9       }
10
11      weights = np.array([cont['weight'] for cont in continents.values()])
12      continent_choices = np.random.choice(len(continents), size=n_samples, p=weights)
13      rotation_vectors = np.zeros((n_samples, 3))
14
15      continent_list = list(continents.values())
16      for i in range(n_samples):
17          continent = continent_list[continent_choices[i]]
18          theta = np.pi/2 - np.deg2rad(continent['lat'])
19          phi = np.deg2rad(continent['lon'])
20
21          theta += np.random.normal(0, 0.2)
22          phi += np.random.normal(0, 0.2)
23
24          axis = np.array([
25              np.sin(theta) * np.cos(phi),
26              np.sin(theta) * np.sin(phi),
27              np.cos(theta)
28          ])
29
30          magnitude = vonmises.rvs(magnitude_kappa, loc=np.pi/4)
31
32          rotation_vectors[i] = axis * magnitude
33
34      return rotation_vectors, continent_choices
```

*Listing 1.* Code for generating the dataset

## F. Additional Numerical Results for Text-Image Generation

**CLIP Similarity**   We generate 5000 samples and evaluate the CLIP similarity between the text and image. For this evaluation, we use CLIP-ViT-large-patch14 and we limit the captions to 77 tokens. We use our sampling default values during this task. We obtain a CLIP score of **18.46** for text-to-image generation, **17.44** for image-to-text generation, and **17.57** for image-text joint generation.

**Generated examples visualization**   We display non-cherry-picking generated examples in all three scenarios in Fig. 8, Fig. 9, Fig. 10.

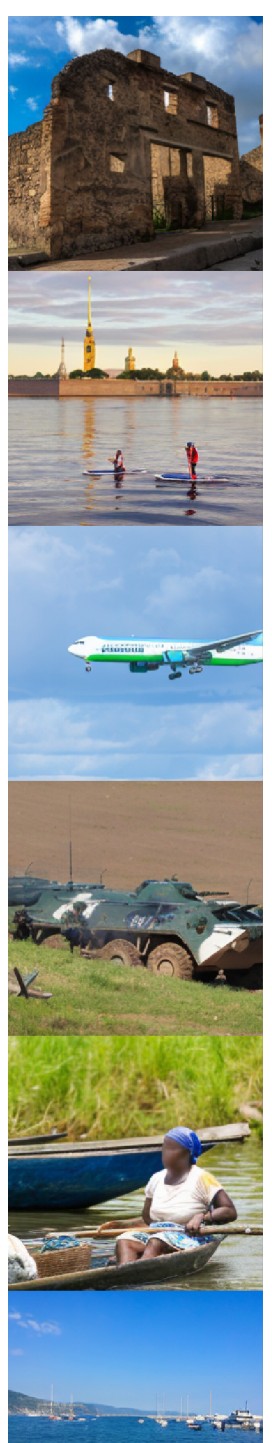

the image features an old , old building with a tall brick wall and a stone wall . the building is situated brick building a surrounded by a lush green field . the stone wall is located near one another , and there are two trees nearby . the style of the image is black and white , which gives it a timeless and classic feel . the black and white color palette adds a sense of nostalgia and elegance to the scene , emphasizing the architectural details and the natural landscape .

the image depicts a scene of several people on surfboards working in a large body of water , surrounded by boats on surfboards . the fishing players soaring through the water , creating a enjoyable waterway to board wings against the sun . the style of the image is a black and white photo , which adds a timeless and classic feel to the scene . the black and white color scheme also emphasizes the contrast between the players ' actions and the beach , drawing attention to the dynamic nature and their interactions being out . subjects

the image features a large green commercial airplane , possibly a jumbo jet , flying through the sky . the airplane is seen from a low angle , emphacapturing its unique size . its bird is filled with clouds . the sky is blue , and the airplane is flying against the clouds . the style of the image is artistic and visually appealing , with the combination of the airplane 's shape and color scheme against the sky of the surrounding clouds creating a visually striking and evocative scene . the image is likely taken with its aviation photography , showcasing the beauty and elegance of skyline , botanical vase airplane that observe be

the image features a group of soldiers in military fatigues , driving down a grassy field . they are wearing cars and other weapons , which are intricately designed displays vehicles . the style of the image is likely to be captured in black and white , which gives it a classic and timeless feel . the soldiers are all positioned in a row , showcasing their guide by their teamwork and unity . the image captures the essence of a military parade , where they come together to share their touch with each another .

the image features a young boy sitting in a boat on a body of water , such as a river or a lush green shoreline and holding a long wooden stick . the boy is wearing a black shirt and jeans , and he appears to be sitting on the bench . the style of the image is a black and white photo , which gives it a classic and timeless feel . the composition of the photo adds a sense of nostalgia and emphasizes the focus on the boy focusing king on rally on the boat as he interlabel ation with the surrounding environment . contempl

the image features a harbor filled with boats docked at the pier , with a sandy beach nearby . it is lined with boats , including a row of small boats with a cannon . the lighthouse 's interest was grand attention to the water water crystal clear , making it visually appealing if it has been interacting with it . the beach appears to be unpaved , with a few potted plants around scattered rocks adding a touch of natural elements to the scene . the sky is pleasant , and the overall atmosphere of the image , creating a mix of natural beauty and serene activities .

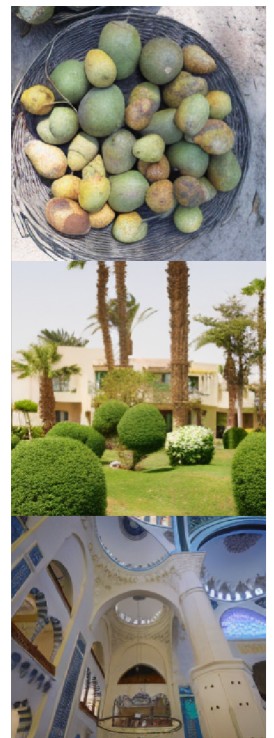

the image features a large pile of fresh produce , including variety of waterlike fruits and vegetables , arranged on top of each other on a table . the handmade fruits are displayed in a visually appealing manner , with different mix of colors and natural textures . the fruits are well - maintained and appear for , creating an aesthetically pleasing and inviting scene . the stand likely to be used to provide protection from the elements and maintain a contemporary look , help the sunlight enhance its appeal and an engaging visual experience .

the image features a lush green field with a large fountain in middle of surrounded by greenery , including flowers and trees . the park is situated in front of a stone wall , and there is a bench in the center of the grassy field . the style of the image is a black and white photo , which gives it and classic and timeless appearance . the contrast between the vibrant green surroundings and sandy fountain , along with the black and white style creates a visually appealing and artistic representation of a park .

the image features a large , ornately natartistic designed mosque with a gold - scheme , domed ceiling and , decorative tiwall designed intricate features details . the mosque is adorned with decorations , including a large chandelier on the mosaic , which adds to the overall grandeur of the space . the style of the mosque is a blend of traditional and modern levels elements , contemporary design . the atmosphere of the image is one of reverence and spirituality , reflecting the rich and history cultural nature of the mosque .

the image features a group of people in a canoe , floating down a river near a river or a lake . the boat is filled with passengers , and there are several people on both the riverbank . the scene is captured in black and white , giving image a vintage or old - fashioned appearance . the people in the raft are wearing traditional clothing , and some of them are wearing a hooded , suggesting that the weather trip is not to the everyday life matter . the overall style of the image is a blend , realistic , and nostalgic atmosphere , capturing could captures the essence of a leisurely ride

the image features a colorful scene with a row of buildings situated along the waterfront , each with a pink door including , there several buildings situated along a boat . the buildings have pink and white trim , which adds to the vibrant and cohesive appearance of overall appearance . the style of the image is likely a combination of minimalism - moment painting and vibrant perspective , which gives it a unique and artistic appearance . monochromatic style further enhances the visual appeal of the scene , capturing the essence of the buildings and their surroundings in the otherwise monochromatic table . the photo used artist 's choice adds depth and interest

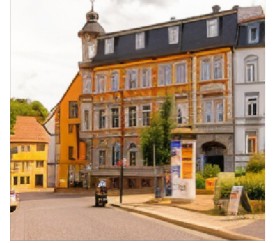

the image features a large , old brick building with a tall clock tower , situated next to a city street . the building has a unique intricate architectural design , featuring a tall steeple and a clock tower . the steepstation is surrounded by street lamps and a bench can placed along the sidewalk . the scene is set outdoors , with sunlight illuminating the area , illuminating the surroundings . the image is in black and white , which adds a sense of timelessness and classic train . the overall composition of the image showcases the architectural beauty of the old and new , showcasing the beauty and charm of the cityscape

*Figure 8.* Visualization of texts generated conditioning on the images.

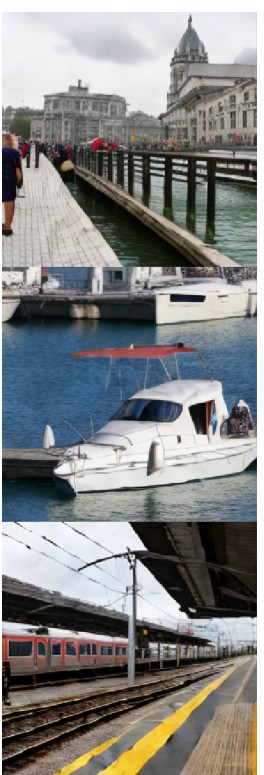

the image features a long pier or bridge filled with people , some of whom are holding umbrellas . the scene takes place in a city , with a large building visible in the background . the style of the image is black and white , which gives it a classic and timeless feel . the composition of the image , with the people and the cityscape , creates a sense of movement and activity . the black and white color palette adds a sense of nostalgia and evokes a feeling of the past , while the people and the cityscape convey a sense of modern urban life .

the image features a white boat floating on a large body of water , such as a lake or ocean , with a person standing on it . the boat appears to be a small motorboat , possibly a speedboat , as it is described as a " speed boat " in the image . the boat is docked at a marina , and there are several other boats in the vicinity . the style of the image is a black and white photo , which gives it a classic and timeless appearance . the focus of the image is on the boat and the person standing on it , capturing the essence of a

the image features a train station with a long platform and a train parked at the end of it . the train is a silver and red commuter train , and it is surrounded by a yellow and black striped platform . the scene is set in a city , with a gray sky overhead . the train station appears to be a busy and bustling location , with several people walking around and a few cars parked nearby . the style of the image is a black and white photo , which gives it a classic and timeless feel . the composition of the image , with the train and the platform as the main subjects

the image features a mountain road at night , with a winding road and a lit - up road sign . the road is surrounded by a forest , and the sky is dark , creating a dramatic and visually appealing atmosphere . the style of the image is artistic and captures the essence of the winding mountain road at night , with the focus on the road , the surrounding forest , and the lit - up road sign . the darkness of the sky and the contrast between the lit - up road and the surrounding darkness create a captivating and visually striking scene .

the image features a city street scene with a brick sidewalk , a bike parked on the side , and a bench . there are several people walking on the sidewalk , and a bicycle is also present . the street appears to be lined with buildings , and there are several potted plants and a tree on the sidewalk , adding to the charm of the scene . the style of the image is a black and white photograph , which gives it a timeless and classic feel . the composition of the image , with the people , the bike , and the potted plants , creates a sense of everyday urban life and the simple

the image features a large , open , paved area with a tall metal pole in the middle . the pole is topped with a red cross , which is a symbol of remembrance . the area is surrounded by trees , giving it a serene and peaceful atmosphere . the style of the image is black and white , which adds a timeless and classic feel to the scene . the black and white color scheme also emphasizes the contrast between the pole and the surrounding trees , drawing attention to the memorial and its significance .

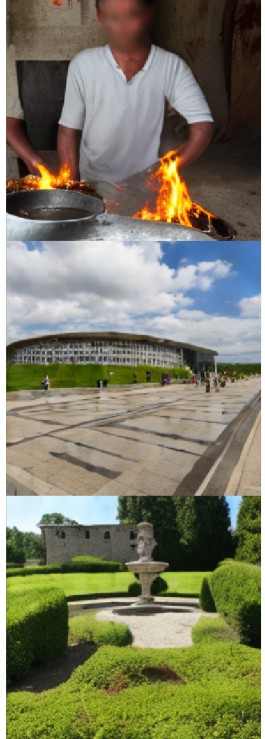

the image features a man standing in front of a large pot filled with food , which is cooking on a stove . the man is wearing a white shirt and a black hat , and he is surrounded by various cooking utensils , such as a spoon and a bowl . the scene is set in a kitchen , and the style of the image is black and white , giving it a classic and timeless feel . the man 's facial expression and the overall composition of the image create a sense of warmth and familiarity , as if the viewer is witnessing a cherished family tradition or a cherished memory .

the image features a large , open , and grassy area with a large building in the background . the building appears to be a stadium or a large event venue , possibly a sports arena or a convention center . the area is filled with people , some of whom are walking around , while others are sitting on benches . the style of the image is in black and white , which gives it a classic and timeless feel . the black and white color scheme adds a sense of nostalgia and elegance to the scene , emphasizing the architectural elements of the large building and the people in the area .

the image features a lush green garden with a fountain in the middle , surrounded by a variety of plants and bushes . the garden is located near a stone building , which could be a house or a historical site . the garden is well - maintained , with neatly trimmed bushes and plants , creating a serene and inviting atmosphere . the style of the image is a black and white photo , which adds a timeless and classic touch to the scene . the combination of the greenery , the fountain , and the stone building creates a harmonious and visually appealing environment .

the image is a black and white photograph of a city street with a tall building in the background . the style of the image is reminiscent of classic or vintage photography , which is characterized by its monochromatic color scheme and the use of contrast to emphasize the shapes , textures , and lines within the scene . the black and white color palette creates a timeless and classic atmosphere , allowing the viewer to focus on the architectural details , such as the tall building and the street , without the distraction of color . this style of photography is often used to capture the essence of a cityscape or to evoke

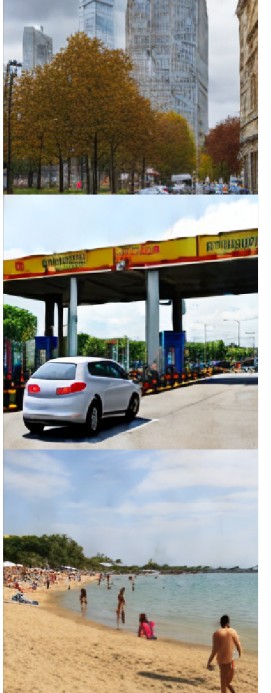

the image depicts a busy highway intersection with several cars and trucks stopped at a toll booth . the toll booth is a large structure with a blue and white color scheme , which stands out against the backdrop of the sky . there are multiple cars and trucks stopped at the toll booth , and the traffic lights are also visible in the scene . the image is in black and white , giving it a classic and timeless feel . the style of the image is reminiscent of classic black and white photography , which emphasizes the contrast between light and shadow , and the overall composition of the scene .

the image depicts a beach scene with a sandy shoreline , where a group of people is enjoying their time . there are several individuals present , including a man and a boy , who are playing in the water . the beach appears to be crowded , with many people engaging in various activities such as sunbathing , swimming , and socializing . the image is in black and white , which gives it a classic and timeless feel . the overall style of the image is nostalgic and evokes a sense of leisure and relaxation , as it captures the essence of a typical beach day .

*Figure 9.* Visualization of images generated conditioning on the text caption.

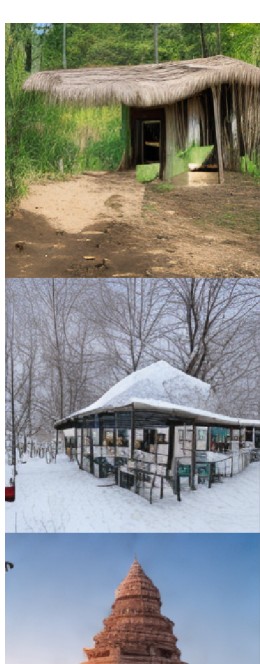

the image features a small green hut or brightness shelter with a door , situated in a lush green field . the hut is surrounded by a variety of trees , huts , which are for commuting conies . the hut has a natural environment . the setting is designed to resemble a village or a traditional asian village , with a peaceful and natural setting . the image is in black and white , which adds a timeless and classic feel to the scene . the style of the image is artistic and evocative , capturing the essence traditional , rural lifestyle and the simple hut within the natural environment .

the image features a snow - covered field with a large , open - covered shelter for visitors . the shelter is surrounded by a fence , and there are several tables and chairs set up outside the shelter . the scene is set in winter colors , suggesting that the image was taken during the winter season . the snow covered landscape and the surrounding trees create a serene and picturesque atmosphere , making it stand out from typical winter snowy environment . the image captures a moment of winter , with people enjoying the cold weather , showcasing the shapes and textures of the snowy landscape .

the image features a large , ornate , and ancient - looking building , resembling a temple or a palace . the building is situated on a city street and is surrounded by a brick wall . the building 's architecture very detailed and intricate , suggesting that it might be a significant cultural landmark or a holds site - colored importance . the use of gold - colored roof and intricate patterns adds to the building 's authenticity and grand appearance . the image is taken during the day , with the sun casting a warm glow on the scene .

the image features a street sign mounted on a pole , which is located in a park square . the sign is written in black italian , indicating the park is a place for people to learn their park . the sign is placed in front of a tree , which adds a classic element to the park . the style of the image is black and white , which gives it a timeless and classic feel . the monochromatic color scheme adds a sense of nostalgia and highlights the contrast between the cat city square and the humorous bicycle rider . the sign is positioned in the park setting , creating a sense of tranquility

the image features a man wearing a blue hat and a red hat , walking down a storefront . he is holding a book , possibly a music , and appears to be reading or walking down the street . the scene is set in a european city , with the man dressed in casual clothing , and the storefront is surrounded by several signs and , which might provide information . the overall style of the image is a black and white photograph , which gives it a classic and timeless feel . the choice of black and white adds composition adds depth and emotion to the scene , making it more engaging and relatable

the image features a large , illuminated christmas tree display , surrounded by lights and decorations . the tree is in a public square , and it is surrounded by a city skyline visible in the background . the tree is made of lights and there are multiple christmas trees , which create a festive atmosphere and a bustling atmosphere . the image is in black and white , which gives it a classic and timeless feel , emphasizing the elegance of the scene .

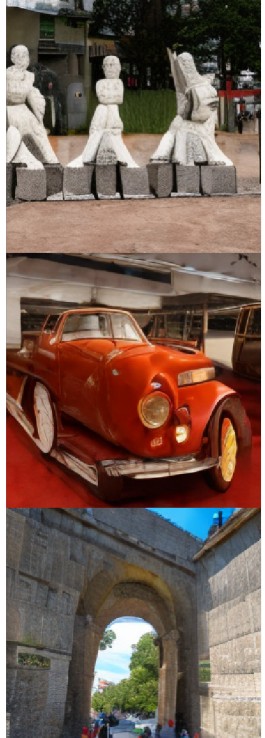

the image features a row of statues or adults holding up hands which are placed on a dirt road in a city . the statues are positioned in a way that suggests they are arranged in a row , making they easily accessible and intricate display . the statues are made of stone and have a white color scheme , which contrasts with the dirt surroundings , drawing attention . the style of the statues is likely a by a traditional art figure , as they are part of a culture or event . the image captures the beauty and uniqueness of the human hands and statues , showcasing their artistic significance and the skill

the image features a vintage orange car parked in a room , which is connected to both played by a large airplane . the cars are parked on a red carpet and surrounded by trees , giving the room a a forest or the setting . the car appears to be a historical vehicle , possibly a museum or a large airplane , and a well - maintained interior . the car 's design and style suggest that these antique vehicles are characterized by a lessness of nostalgia and sophistication . the scene is set in a hall with a modern style that natural and well - lit highlights the vehicles in a stylized and

the image features a large stone archway with a stone wall , which is located at a park . the archway is filled with people , some of whom are walking around the area . the scene is set on a sunny day , with the sunlight streaming in the sunlight illuminating the scene . the style of the image is a black and white photo , which adds a timeless and classic feel to the scene . the composition of the image emphasizes the grand archway and the natural beauty of the sunlight , creating a stone appealing and memorable experience .

the image depicts a bustling outdoor market market , where several people are walking around , possibly shopping for vegetables . there are several individuals in the scene , interacting with each other , and shopping , and some of them are lying on the ground . the market filled with various items , such as vegetables , which suggests that they contribute to the lively atmosphere and inspire people to compete and tied . the image is in black and white , giving it a classic and timeless appearance . the people in the image are engaged in discussing the market 's produce , possibly discussing or examining them fresh , purchasing items , or simply

the image depicts a busy street scene with a large crowd of people gathered around , market with a variety of items for selling flowers . the market is situated on a brick road , giving it a quaint and charming atmosphere . the vendors are filled with different of of flower items , including fruits , vegetables , and flowers on display . the market is bustling with activity , both the place and shot multiple shoppers . the overall style of the image is a close - up , focusing on the details and the abundance and interactions that take place in the world .

the image features a large statue of a person , possibly a man , standing on a tall in the shade over a forest . the statue is surrounded by trees , creating a serene and picturesque setting . the statue is made of intricate carvings and appears to be a part of a historical building , as it is described as a " statue ." the style of the image is artistic and evocative , capturing the beauty and composition . the mythical nature of the statue and the surrounding environment create a visually appealing and captivating atmosphere , inviting the viewer to appreciate the details and majesty of the scene .

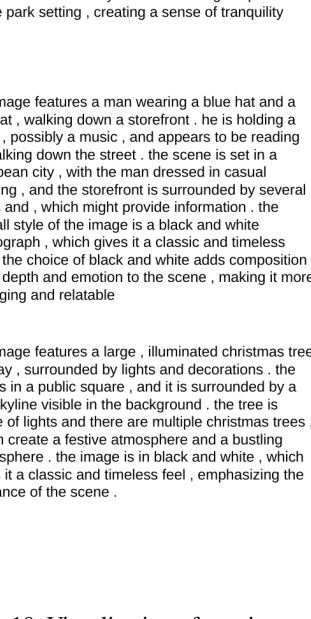

*Figure 10.* Visualization of text-image pairs generated jointly and unconditionally.

