# OpenReview forum: "Diffuse Everything: Multimodal Diffusion Models on Arbitrary State Spaces"
_ICML.cc/2025/Conference — ICML 2025 poster_

### Official Review · Reviewer_R8im · 2025-03-10

**Overall Recommendation:** 3

**Summary:**

- The paper introduces a framework for multimodal diffusion models on arbitrary state spaces via independent noise schedules for each modality.
- In particular, the paper proposes a theoretically grounded framework for multimodal diffusion of continuous as well as discrete state spaces.
- After training, the diffusion model can be used for unconditional joint generation of multiple modalities or a single modality conditioned on another one.
- An evaluation for text-to-image generation using a multi-modal DiT (MMDiT) architecture and mixed-tabular data synthesis shows competetitive performance compared to baselines despite smaller model sizes.

**Claims And Evidence:**

All claims are supported by convincing evidence except for:
- The authors claim that on text-to-image generation they "achieve similar performance as commercial-grade models using a small model without leveraging powerful extra encoders" (lines 100 ff., right column).
  - The perfomance compared to MMDiT-improved with the same number of training images and a similar-sized model or PixArt-alpha XL/2 with also a similar-sized model but double the number of training images is significantly worse.
  - It is unclear how the proposed method does not use extra encoders.

**Essential References Not Discussed:**

There is an essential related work that has not been discussed:
- UniDiffuser [1] proposes a multi-modal transformer-based diffusion model with independent noise levels / timesteps for each modality with applications to text-to-image, image-to-text, and joint text- & image-generation as well as discussion of enabling classifier-free guidance. Therefore, the core idea of this paper is not novel anymore.

[1] One Transformer Fits All Distributions in Multi-Modal Diffusion at Scale. ICML 2024

**Experimental Designs Or Analyses:**

All experimental designs and analyses seem to be valid.

**Methods And Evaluation Criteria:**

The proposed methods make sense except for:
- The paper claims that it does not make use of any extra encoders (also blank cell in table 1).
  - I do not understand how this is done. Does it mean that diffusion of images is done on pixel level? But then the authors mention "noisy latents" in line 310, right column and the MMDiT architecture is not built for pixel-space diffusion.  Appendix B.2 mentions a joint embedding and continuous decoder trained for text-image alignment but this whole architecture remains unclear.

The used evaluation criteria follow standard setups and make sense.

**Other Comments Or Suggestions:**

No other comments or suggestions

**Other Strengths And Weaknesses:**

Strengths:
- The paper introduction motivates the advantages of continuous and discrete diffusion and the potential of multi-modal diffusion well.
- The proposed noisy guidance in section 3.4. renders an additional interesting technical contribution.
- The flexibility of the framework enabling unconditional multimodal generation as well as conditional single-modality generation given other modalities is elegant.
- The experimental results for tabular data synthesis using a much smaller model than baselines are promissing.

Weaknesses:
- Lack of clarity:
  - The introduction states that VAEs are mainly used for mapping different modalities into a single modality for diffusion. However, the main motivation of using VAEs is to have a compressed latent space for more efficient diffusion model training as introduced by the original latent diffusion paper.
  - In figure 3, the zoom-in seems to contradict with the rest of the plot.
  - As stated above, it remains unclear how the proposed method does not make use of any extra encoders.
  - A short description of the metrics used for evaluation of tabular data synthesis would be helpful.
  - It is unclear why the proposed approach is better than all baselines for tabular data synthesis despite using a so much (100 to 200 times) smaller model. What are the key differences?
- Section 3.1. is not very clear about whether the unified perspective is from prior work, in which case it should be part of preliminaries, or whether it is something novel.
- The used MMDiT architecture as outlined in figure 2 is largely inspired by StableDiffusion 3 and therefore it should be cited in the figure and the architecture description (first paragraph of section 4.1).

**Questions For Authors:**

1. Could you comment on the main differences compared to UniDiffuser (see review section Essential References Not Discussed)?
  - Since this work basically proposed the same approach, a discussion of the key differences is necessary to understand the novelty of this paper.
2. How exactly does the used model not rely on extra encoders? Could you please specify the architecture (joint embedding and continuous decoder mentioned in appendix) and how all components are connected?
  - This is essential to resolve the lack of clarity w.r.t. one of the key differences compared to baselines.
3. It is unclear why the proposed approach is better than all baselines for tabular data synthesis despite using a so much (100 to 200 times) smaller model. What are the key differences?

**Relation To Broader Scientific Literature:**

There has been a lot of research on diffusion models for continuous spaces in recent years. Recently, diffusion models for discrete spaces gained attention. There have papers proposing unified perspectives like denoising Markov models [1] as mentioned by this paper in lines 149, right column. This paper focuses on the formal derivation of multimodal diffusion with individual noise levels for each modality. Experiments for text-to-image generation leverage the MMDiT architecture introduced by previous work StableDiffusion 3 [2].

- [1] From Denoising Diffusions to Denoising Markov Models. ournal of the Royal Statistical Society Series B: Statistical Methodology. 2024
- [2] Scaling Rectified Flow Transformers for High-Resolution Image Synthesis. ICML 2024

**Theoretical Claims:**

I did not check the proofs of all theoretical claims provided in Appendix A.

---

> ### Author Rebuttal · Authors · 2025-04-01
>
> We thank the reviewer for their time and valuable insights.
> ### Regarding T2I performance
>
> We want to emphasize that our work focuses on training for the next generation of multimodal diffusion models for **multiple tasks** rather than a single one, which in general is much more challenging. Despite the smaller model, when compared to other **multi task** models like Chameleon or JetFormer (which are commerical-grade), we can still achieve better results on the MS-COCO text to image task.
>
> The comparison to T2I models such as MMDiT-improved and PixArt-alpha is included to provide a view of **single task** SOTA models. A direct comparison is unfair due to the huge difference in task difficulty. Despite this, we achieve a decent T2I generation quality, indicated both by the FID score and a high visual quality (samples in Appendix D).
> ### Regarding VAEs
>
> We apologize about the confusion. We agree that in image diffusion models VAEs are used to get a compressed latent space. We were trying to state that in the case of multimodal models, encoders are used to set data into the same space with techniques such as VQ-VAE or VQ-GAN. For example, images can be tokenized, changing from a continuous state space to a discrete state space, where we get compatibility with the text modality. In our work we would like to present an approach that can treat each data modality in its native state space.
>
> ### Comparison to UniDiffuser
>
> We thank the reviewer for pointing us to this reference, we will include it in the revision. UniDiffuser uses the similar idea of introducing separate noise levels for different modalities. However they encode the continuous image and discrete text into continuous embeddings. They then train a continuous diffusion model to generate the latent emmbeddings. After using proper decoders they can recover the image and text. This technique is of the kind defined in the previous point, where both modalities are moved into the same state space and the generation is performed there.
>
> Different from UniDiffuser, we propose treating each modality in its native space. We treat images as a continuous object and text as a discrete one. We do so by leveraging advances in continuous and discrete diffusion and introducing a novel framework that allows combining different diffusion models together.
> ### Regarding errors on Zoom-in plot
>
> We sincerely apologize for the confusion and thank you for catching this. Indeed, the colors of the zoom-in are incorrect, there should be a change of colors with the $w=3$ and $w=5$ curves. We will correct this in further revisions.
> ### Regarding use of extra encoders
>
> Here we meant the encoders to change data from one space to another as described above. As well as encoders like CLIP which are usually used in T2I diffusion models. Which encode text into a latent vector which is used to condition the diffusion model. Instead, we directly apply a learnable embedding to the text (with nn.Embedding) and learn its own hidden representations.
>
> We apologize since we had a typo in our table, we did not train in the pixel space. We used the usual stable diffusion VAE to train the continuous component in the latent space (for dimension compression). As expressed in other parts of our paper (as you described), we trained in the latent space for images.
> ### Regarding tabular evaluation metrics
>
> The metrics used in tabular data experiments are described in Appendix C.3. For better readability, we will also add more descriptions of these metrics in the main text in the updated version.
> ### Regarding improvement on tabular data
>
> We believe that the reduction in model size is due to two major reasons:
> 1) We operate on the native state space for mixed type tabular data and don't use any complicated encoders for embedding each modality. This reduces the model size as encoders, such as VAE, can be parameter-heavy.
> 2) Based on point 1, we designed a new score network based on transformers that directly takes a mixed-type tabular data point as input. Our score network adopts the idea of early-fusion (also discussed and adopted in Chameleon), which learns a joint embedding between modalities starting from the first attention layer. We suspect that early-fusion approaches are more effective and parameter-efficient.
> We will add more discussion related to this in the updated version for better readability.
>
> ### Regarding Section 3.1
>
> Section 3.1 discusses a statistical and theoretical framework of diffusion models by drawing connections to generalized score-matching objectives. While the connection is proposed in prior work (which we have clearly indicated), we leverage this perspective to derive a framework for multi-modal diffusion models of both practical and theoretical importance. For ease of understanding and notation consistency, we kept this content in Section 3.
>
> ### Regarding reference to MM-DiT architecture plot
>
> We will gladly add the citation as described.

---

> > ### Comment · Reviewer_R8im · 2025-04-04
> >
> > Thank you for your rebuttal. It addresses some of my concerns, especially the lack of clarity w.r.t. the role of encoders.
> > Given that the idea of separate noise levels for different modalities was already proposed by UniDiffuser and is therefore not novel anymore, I would like to better understand the remaining novelty of this paper, i.e., the mixture of diffusion for different (possibly non-continuous) spaces for different modalities. Please correct me, if I am missing further novel contributions.
> > To that end, I have additional questions regarding the derived training objective.
> >
> > Is it significantly different from a simple (weighted) sum of the training objectives for the different modalities?
> > - If yes, why would a weighted sum be incorrect, i.e., just training a shared model to optimize both objectives jointly?
> > - If no, why is it non-trivial that the joint training objective is a weighted sum of the individual ones?
> >
> > Thank you very much in advance!

---

> > > ### Author Response · Authors · 2025-04-07
> > >
> > > We thank the reviewer for replying to our rebuttal and the scholarship of being willing to discuss further. The answer to your important question is, no, and we will now explain why it is a nontrivial result even though the joint training objective is a weighted sum of the individual ones. There will be three points (A1-A3) in our answer. Afterward, two other contributions of this work will also be mentioned (B & C). Overall, your question really helps us better outline our contributions and they will be made clearer in a revision.
> > >
> > > **A)** It is indeed not the first time that people have considered continuous + discrete modalities, but this time we proposed a general theoretical framework of multimodal diffusion model. It leads to multiple new and useful results such as the following:
> > >
> > > **A1)** Consider using a joint training objective that sums up objectives for each modality. Each term (for each modality) can use multiple choices of its loss function (e.g., Bregman Divergences provide loss functions by picking a convex function $F$), so how to make good (combinatoric) choices? It has been shown in [1] that for each modality there is a specific loss function (call it $i$) that produces the best results. (For instance in the Euclidean case we use $F = |\cdot|^2$, but in the discrete case we use $F=\sum_i p_i (\log p_i - 1)$ while other options perform poorly). Other modalities might have their own loss function (call it $j$). Our theoretical framework automatically gives pairs of $i$ and $j$ that work well, which simplifies practical decisions.
> > >
> > > This is not only verified through empirical observations but there are provable connections between our training objective and statistical properties like KL divergence and ELBO. We will expand on such properties in the revision.
> > >
> > > **A2)** When adding terms up, it is not so clear a priori which one of the following two should be used (continuous+discrete modalities are chosen as an example for readers' familiarity):
> > > *version 1*
> > > $$\mathbb{E} \bigg[\|s^X-\nabla_x \log p(x_t, y_s, t,s | x_0, y_0) \|^2+\sum_{z\neq y} \big( s^Y_z - \frac{p(x_t, z,t,s | x_0, y_0)}{p(x_t, y_s, t,s | x_0, y_0)} \log s^Y_z\big)\bigg] $$
> > > *version 2*
> > > $$\mathbb{E}\bigg[\|s^X-\nabla_x\log p(x_t,t|x_0)\|^2
> > > +\sum_{z\neq y}\big(s^Y_z-\frac{p(z,s|y_0)}{p(y_s,s|y_0)}\log s^Y_z\big)\bigg]$$
> > > Where $s^X = s_\theta^X(x_t,y_s,t,s)$, $s^Y_z = s_{\theta}^{Y}(x_t, y_s, t,s)_z$.
> > >
> > > We used *version 2* and, very importantly, can prove that not only are both versions equivalent, but also they are equivalent term by term. Additionally, we showed that such a denoising version will exist for other modalities in thm 3.2.
> > >
> > > We can prove this result using the nice properties of the score function (among various state spaces) as well as the decoupled forward processes. Since $p(x_t,y_s,t,s|x_0,y_0) = p(x_t, t | x_0) p(y_s,s|y_0)$, this implies that
> > > $$\nabla_x\log p(x_t,y_s,t,s|x_0,y_0)= \nabla_x\log p(x_t,t|x_0)$$$$\frac{p(x_t, z,t,s|x_0,y_0)}{p(x_t, y_s, t,s|x_0,y_0)} =\frac{p(z, s| y_0) }{p(y_s, s| y_0) }$$
> > > It becomes apparent that the term by term equality is a consequence of score functions being independent of their normalizing constant. Therefore, a result like this one wouldn't be true in general for other classes of loss functions. For this reason, it was not expected that adding the **unimodal** losses would give an objective that recovers the **multimodal joint marginal**.
> > >
> > > **A3)** Thanks to the general framework, we can easily do all the above for other modalities as well (not necessarily continuous + discrete). We added an experiment on our rebuttal to reviewer QDhM under the title "Regarding the evaluation on other domains" where we generated a toy example of Riemannian + discrete data.
> > >
> > > **B)** It is surprising, but in fact, so far we are unaware of any proof that the joint score learned from *version 2* will enable the backward process to sample from the true data distribution. We proved this in Appendix A.3. Such calculation is nontrivial and it is crucial. Without this result, score learning would be meaningless as there's no relation to generative modeling.
> > >
> > > **C)** Even if the joint training objective appears simple due to being a sum, its practical optimization is more difficult than all its unimodality components. In Appendix B.2. we proposed a strategy to make training more tractable and it was effective. This point was demonstrated in an ablation study shared with Reviewer g74r under the title "Regarding architecture choice for text-image", where naively training with the proposed loss, without using our proposed strategy, led to terrible performance (FID $73$, CLIP score $9$) while our training strategy can achieve competitive performance (FID $16$, CLIP score $18$). We think this engineering trick is also a contribution.
> > >
> > > We sincerely hope our explanations can earn your (re)consideration, but regardless, thank you for helping us significantly improve our presentation.
> > >
> > > ### Refs
> > > [1] 2310.16834

---

### Official Review · Reviewer_g74r · 2025-03-12

**Overall Recommendation:** 2

**Summary:**

This paper proposes a novel diffusion-based framework for both continuous and discrete multimodal data (images and text). Specifically, for continuous image modality, the paper utilizes diffusion process as forward and backward process. For the discrete text modality, the paper uses CTMC to determine the state of text tokens. The two modalities are unified through a single diffusion process, but using different forward and backward pass formulas, with the allowance of using an asynchronous timestep. For experiments, the authors evaluate the performance of the model on trend, efficiency, shape, precision, and recall. The proposed method achieves competitive results with other models, while using a much smaller model size.

**Claims And Evidence:**

The claims in this paper, including competitive performance on text-image generation and mixed-type tabular data synthesis are supported by experiments on corresponding evaluation datasets.

**Essential References Not Discussed:**

Related works are well-discussed.

**Experimental Designs Or Analyses:**

The experiments on text-image generation and mixed-type tabular data synthesis are valid, but the performance is not superior than other models.

**Methods And Evaluation Criteria:**

The evaluation benchmark and baselines seems insufficient here. Since this work is in the same line of work with Transfusion, Show-o, JanusPro, etc., it would be useful to do evaluation on GenEval, DPG-Bench for text-to-image generation, and POPE, MME-P, MMB, SEED, GQA, MMMU, MM-Vet, etc. for image understanding.

**Other Comments Or Suggestions:**

NA

**Other Strengths And Weaknesses:**

Major Strength:

- The idea of creating a unified model that uses discrete diffusion for text and continuous diffusion for images is interesting.


Major Weakness:

- Limited technical novelty: The overall idea of this paper is very similar with the literature for protein prediction [1], which also uses a unified diffusion model for discrete and continuous tokens. Specifically, both work uses diffusion process for continuous tokens, and CTMC for discrete tokens. Ideas including asynchronous timestep is also first proposed in that paper. In addition, the authors also uses MM-DiT block proposed in Stable Diffusion 3. Therefore, the overall novelty in this paper seems limited.

- Weak result for image generation: The FID score in Table 1 is worse than most baselines, including MMDiT-improved, which has been trained on the same number of images and with comparable trainable parameters. Therefore, the the text-to-image generation capability is not proved effective.

[1] Campbell, A., Yim, J., Barzilay, R., Rainforth, T., and Jaakkola, T. Generative flows on discrete state-spaces: Enabling multimodal flows with applications to protein co-design. arXiv preprint arXiv:2402.04997, 2024.

**Questions For Authors:**

Some questions about missing ablation experiments:

- Why not apply MM-DiT for all blocks? Ablation experiments on model architecture would be helpful.

- The paper didn't use text encoder, but encode the text into learnable embeddings directly. Could the authors provide some comparison experiments about using and not using text encoder?

**Relation To Broader Scientific Literature:**

This work is related to unified model for multimodal generation and understanding, including Show-o, Janus series, Transfusion, etc. These prior works mainly focus on designing a unified model architecture for modality fusion, while usually apply separate training objectives for text and images. In contrast, this work focus on exploring using a unified diffusion objective for both modalities.

**Theoretical Claims:**

I didn't check the theoretical proofs carefully.

---

> ### Author Rebuttal · Authors · 2025-04-01
>
> We thank the reviewer for their time and valuable insights.
>
> ### Regarding evaluation benchmarks
>
> We'd like to emphasize that the objective of our work is to introduce a general framework for training multimodal generative models using diffusion, as opposed to being considered a task specific method.
>
> Our work is slightly different from Transfusion, Show-O, etc. These models are capable of multimodal understanding through instruction finetuning, whereas our model does not. Therefore, we do not evaluate on benchmarks that requires understanding.
>
> As for the text-to-image generation benchmark, while GenEval, DPG-Bench are also popular choices, we found that MS-COCO is more suitable since it has been selected for evaluation for all most all models with parameters ranging from million to billion levels. We want to emphasize that one major difference between our models and most of the others on the table is that our is small while others is usually of billion order parameters. To accommodate for the parameter differences, we eventually selected MS-COCO and do not go with other larger benchmarks.
> ### Regarding T2I performance
>
> We want to first emphasize that our work focuses on training for the next generation of multimodal diffusion models for **multiple tasks** rather than a single one, which in general is much more challenging. Despite the small model size we use, when compared to other **multi task** models (with bigger size and more computational resources) like Chameleon or JetFormer, we can still achieve better results on the MS-COCO text to image task. For this reason we respectfully disagree that the text-to-image generation capability is not proved effective.
>
> Regarding the comparison to other solely T2I models listed in table such as MMDiT-improved, we want to remark that we listed these results to provide a view of top performing **single task** model in the literature. As we have explained before, a direct comparison with MMDiT-improved would be unfair due to the huge difference in task difficulty. Despite the overall more challenging problem, we still achieves a decent T2I generation quality, indicated both by the FID score and a high visual quality (see more examples in Appendix D). Therefore, we believe the achieved T2I result should be understood as a **merit** rather than a weakness.
> ### Regarding Tabular performance
>
> Our approach does show superior performance in multiple tasks on tabular data synthesis than other models while using a **significantly smaller model** as shown in Table 2, which we believe should be acknowledged as a major improvement over previous works.
> ### Regarding technical novelty
>
> It is true that similar ideas have been proposed before as we have acknowledged. These works use ad-hoc concatenations of diffusion models to create powerful generative models. However, these findings are not based on solid theoretical justification as to why this is a valid thing to do. In our work, we present a general framework by extending [2] and using the per-modality generators to get valid loss functions and obtaining valid processes that preserve the distributions, confirming the findings from [1]. Additionally our framework allows not only works like [1] but exploration on other domains with other kinds of data.
> ### Regarding architecture choice for text-image
>
> We adopt this architecture of combined MM-DiT blocks and DiT blocks for the staged training strategy discussed in Appendix B.2. Our architecture design allows an effective implementation of this training strategy. To demonstrate the importance of it, as well as the modular set up of our network, we train a network composed of only MMDiT blocks and train on both tasks at once. This serves as an ablation on the architecture and training strategy. We use a network of similar size and use the same hyperparmaters as in stage 1 in Table 4. This trained model achieves an FID of 73 on the text to image MS-COCO task and a CLIP score of 9.07 on the joint generation set up. This shows that naively using all MM-DiT blocks results in terrible results and that a staged training process is required as demonstrated by our improved results.
> ### Regarding use of additional text encoders
>
> Most text to image diffusion models make use of a CLIP encoder for the text. However our model also receives as input text that has been corrupted by a masking process. CLIP and other text encoders are not trained on text of this form. For this reason we decided to omit the text encoder from our  network and don't have such a baseline.
>
> #### References
>
> [1] Campbell, A., Yim, J., Barzilay, R., Rainforth, T., and Jaakkola, T. Generative flows on discrete state-spaces: Enabling multimodal flows with applications to protein co-design. arXiv preprint arXiv:2402.04997, 2024.
>
> [2] Benton, Joe, et al. "From denoising diffusions to denoising markov models." Journal of the Royal Statistical Society Series B: Statistical Methodology 86.2 (2024): 286-301.

---

### Official Review · Reviewer_QDhM · 2025-03-13

**Overall Recommendation:** 3

**Summary:**

In this paper, the authors focus on the problem of using diffusion to model multi-modal data domains, especially text and image data. To this end, they propose a novel approach to the noise schedule which is distinct for each modality. They justify their approach theoretically with proofs. They then evaluate their approach empirically on joint text-image generation and tabular data domains. They show quantitatively that their approach is an improvement over baselines on the tabular data.

## update after rebuttal
Although some of the reviewers had some concerns, I think this paper has some potential contributions. I keep my assessment of weak accept.

**Claims And Evidence:**

Yes, to some extent. The authors justify their approach on theoretical grounds, and they show promising results on image-text and tabular data. However, it isn't clear if the results are general enough to hold to other multi-modal data domains.

**Essential References Not Discussed:**

No.

**Experimental Designs Or Analyses:**

While the experimental design is sound, the breadth of evaluation is narrow. The proposed approach is quite general, but the only domains that are evaluated are image/text data and tabular domains.

**Methods And Evaluation Criteria:**

The methods are sound, and the evaluation criteria are appropriate.

**Other Comments Or Suggestions:**

None.

**Other Strengths And Weaknesses:**

The approach is novel and shows promise, but the scope of evaluation is limited to specific domains. The argument of the paper could be strengthened with evaluations on other multi-modal domains (such as video + audio).

**Questions For Authors:**

Given the general nature of the approach the choice of evaluation datasets (images/text, tabular data) is quite constrained. Would it be possible to evaluate on other multi-modal domains (i.e. video + audio)?

**Relation To Broader Scientific Literature:**

Compared to prior ideas, the authors propose a novel approach to diffusion on multi-modal data, specifying a noise schedule that is separate for each modality.

**Theoretical Claims:**

The theoretical claims appear to be correct albeit not all details were checked.

---

> ### Author Rebuttal · Authors · 2025-04-01
>
> We thank the reviewer for their time and valuable insights.
> ### Regarding Generality  of the set up
>
> Our work built on the general set up of [1] which includes at least Euclidean, discrete, Riemannian and Wright-Fischer diffusions. We improve upon it by leveraging separate noise levels on different modalities which inmediately allows for combinations of modalities while leveraging joint, single-modality and conditional generation in a single model. For this reason our approach generalizes to at least all processes covered by [1].
> ### Regarding the evaluation on other domains
>
> We must emphasize that our work focuses on introducing a general methodology for dealing with multimodal diffusions. Both the tasks of text+image and tabular data are challenging tasks that require lots of engineering and a single of these tasks is usually the study of a single work. Moreover, the audio-video example is not the best use case to deliver our point as both modalities are represented in continuous values, which in theory can be tackled with a unimodal continuous diffusion model. Therefore, we didn't attempt to do this example.
>
> To further highlight that we are proposing a robust framework of both pratical and theoretical importance, we have included a new example regarding a toy problem that includes Riemannian + Discrete data to demonstrate the versatility of our method in other setups.  This can be found in the following annoymized repo (https://anonymous.4open.science/r/ICML-Rebuttal-FD00/Rebuttal.pdf)
>
> #### References
>
> [1] Benton, Joe, et al. "From denoising diffusions to denoising markov models." Journal of the Royal Statistical Society Series B: Statistical Methodology 86.2 (2024): 286-301.

---

### Decision · Program_Chairs · 2025-05-01

**Decision:**

Accept (poster)

**Comment:**

The authors provide a method and derivations for performing joint diffusion on multiple modalities. The paper received mixed scores after the first review round. The reviewers highlighted the general paradigm and the technical contributions in the framework as positive. Concerns were raised regarding novelty of the individual contributions, clarity and result quality. While most concerns could be addressed in the rebuttal, concerns about novelty remained after discussion with the authors.
Specifically, there is a previous work (One Transformer Fits All Distributions in Multi-Modal Diffusion at Scale. ICML 2024), which the authors missed in their paper and which is a strongly related and crucial work. During the rebuttal, the authors promised to add and discuss it to the draft, and also were able to outline how their method significantly contributes on top of it. In the end, the discussion convinced two out of three reviewers to vote for accepting the paper.

I will follow with a (weak) accept recommendation. I agree that there is novelty in this work and that the technical and theoretical framework can be valuable for future work. At the same time, the overall novelty is slightly dampened by the mentioned publication.
If the paper gets accepted in the end, the authors should add an in-depth discussion of the mentioned previous work to the paper.